# STAT3 promotes RNA polymerase III-directed transcription by controlling the miR-106a-5p/TP73 axis

**Cheng Zhang[1], Shasha Zhao[1]\*, Huan Deng[1]\*, Shihua Zhang[1], Juan Wang[1,2]\*, Xiaoye Song[1], Deen Yu[1], Yue Zhang[1], Wensheng Deng[1]\***

[1]School of Life Science and Health, Wuhan University of Science and Technology, Wuhan, China; [2]School of Materials and Metallurgy, Wuhan University of Science and Technology, Wuhan, China

**Abstracts** Deregulation of Pol III products causes a range of diseases, including neural diseases and cancers. However, the factors and mechanisms that modulate Pol III-directed transcription remain to be found, although massive advances have been achieved. Here, we show that STAT3 positively regulates the activities of Pol III-dependent transcription and cancer cell growth. RNA-seq analysis revealed that STAT3 inhibits the expression of TP73, a member of the p53 family. We found that TP73 is not only required for the regulation of Pol III-directed transcription mediated by STAT3 but also independently suppresses the synthesis of Pol III products. Mechanistically, TP73 can disrupt the assembly of TFIIIB subunits and inhibit their occupancies at Pol III target loci by interacting with TFIIIB subunit TBP. MiR-106a-5p can activate Pol III-directed transcription by targeting the TP73 mRNA 3' UTR to reduce TP 73 expression. We show that STAT3 activates the expression of miR-106a-5p by binding to the miRNA promoter, indicating that the miR-106a-5p links STAT3 with TP73 to regulate Pol III-directed transcription. Collectively, these findings indicate that STAT3 functions as a positive regulator in Pol III-directed transcription by controlling the miR-106a-5p/TP73 axis.

## Editor's evaluation

The author arrived at the convincing conclusion that STAT3 expression promotes TFIIIB assembly through miR-106A-5p-mediated inhibition of TP73 expression, thereby increasing Pol III transcription, which contributes to enhanced cell proliferation. The data are very good and clearly support the proposed model.

## Introduction

Human RNA polymerase III consisting of 17 subunits synthesizes a subset of medium-sized non-coding RNA molecules, including tRNA, 5S rRNA, 7SL RNA, U6 snRNA, and others. Some of these RNA molecules have been well characterized and are essential to a range of cellular activities such as translation (5S rRNA and tRNA), RNA splicing (U6 snRNA), protein translocation (7SL RNA), Pol II transcription elongation (7SK RNA), and so on *Moir and Willis, 2013*; *Turowski and Tollervey, 2016*; *Leśniewska and Boguta, 2017*; *Willis and Moir, 2018*; *Yeganeh and Hernandez, 2020*. Transcription initiation relies on the occupancy of Pol III apparatus at the promoters of Pol III-transcribed genes and can be classified into three initiation modes (types I, II, and III; *Moir and Willis, 2013*; *Turowski and Tollervey, 2016*; *Leśniewska and Boguta, 2017*). During the transcription initiation of tRNA genes (type II), TFIIIC initially binds to the internal elements box-A and box-B within the tRNA-coding region; TFIIIB is subsequently recruited upstream of the transcription start

**\*For correspondence:**
zhaoshasha@wust.edu.cn (SZ);
denghuan@wust.edu.cn (HD);
wangjuan@wust.edu.cn (JW);
dengwensheng@wust.edu.cn
(WD)

**Competing interest:** The authors declare that no competing interests exist.

site (TSS), and Pol III finally binds to the TSS through interacting with TFIIIB (*Moir and Willis, 2013*; *Yeganeh and Hernandez, 2020*). In addition to Pol III general transcription factors, oncogenic factor (c-MYC), tumor suppressor (p53), and signaling transduction factors (JNK and PTEN/AKT) can also tightly modulate Pol III-mediated transcription (*Gomez-Roman et al., 2003*; *Crighton et al., 2003*; *Sriskanthadevan-Pirahas et al., 2018*; *Zhong and Johnson, 2009*; *Woiwode et al., 2008*). MAF1, a well-characterized factor repressing Pol III-directed transcription (*Kulaberoglu et al., 2021*; *Bonhoure et al., 2020*), has recently been shown to regulate osteoblast differentiation and bone mass formation (*Phillips et al., 2022*), and this factor has also been confirmed to improve cardiac hypertrophy and inhibit the synthesis of Pol III products by interacting with extracellular signal-regulated kinase (ERK) (*Sun et al., 2019*). It has been reported that tRNA (tRNA) biogenesis and specific tRNAs such as tRNA$^{Leu}$-CAA and tRNA$^{Tyr}$-GTA are required for the regulation of senescence *Guillon et al., 2021*. These findings suggest that Pol IIII transcription and its products can regulate more biological functions than we thought. Deregulation of Pol III products or the Pol III transcription machinery components closely correlates with carcinogenesis (*White, 2011*). For instance, hepatocellular carcinomas induced by alcohol showed abnormally high expression of BRF1 and Pol III products (*Lei et al., 2017*). Pol III subunit G (POLR3G) silencing inhibits the expression of a small non-coding RNA (snaR-A) that is related to cancer proliferation and metastasis, and POLR3G upregulation is associated with poor survival outcomes in several cancers (*Van Bortle et al., 2022*). In the latest work, GATA4 has been shown to drive Pol III-directed transcription and cancer cell proliferation by enhancing the expression of SP1, which can directly activate the transcription of *Brf1* and *Gtf3c2* genes by binding to their promoters (*Peng et al., 2020*; *Zhang et al., 2022*). Despite enormous advances in Pol III-dependent transcription, the mechanisms and factors governing this process remain to be identified.

Signal transducer and activator of transcription 3 (STAT3), an important effector in the IL-6/JAK/STAT3 signaling pathway, regulates diverse biological processes, including apoptosis, angiogenesis, immunosuppression, cell proliferation, and migration (*Srivastava and DiGiovanni, 2016*; *Johnson et al., 2018*; *Lee et al., 2019*; *Yang et al., 2019b*; *Jin, 2020*). During the activation of canonical STAT3 signaling, elevated IL-6 molecules bind to the IL-6-receptor α and form a complex with glycoprotein p130, which subsequently triggers the activation of JAK/STAT3 signaling (*Srivastava and DiGiovanni, 2016*; *Lee et al., 2019*; *Jin, 2020*). In addition to Janus-activate kinases (JAKS), STAT3 phosphorylation at tyrosine 705 (Y705) is also mediated by other kinases such as ABl and c-Src (*Srivastava and DiGiovanni, 2016*; *Lee et al., 2019*). Upon phosphorylation, STAT3 is dimerized and transported into nuclei through a nuclear localization sequence, where phosphorylated STAT3 dimers modulate the expression of its target genes by binding to STAT3 consensus sequence (TTC[N]$_3$GAA). STAT3 has been shown to regulate the transcription of genes related to survival (BCL-XL and Mcl1), cell cycle (CyclinD1), cell migration (MMP2, MMP9, and Vimentin), and angiogenesis (VEGF, HGF, and HIF1α) and immunosurveillance (NF-kB and CXCL10; *Lee et al., 2019*; *Catlett-Falcone et al., 1999*; *Lin et al., 2011*; *Xu et al., 2005*; *Chen et al., 2015*; *Stumhofer et al., 2007*). Besides, non-canonical STAT3 signaling has been identified to play a role in maintaining mitochondrial functions, and unphosphorylated STAT3 (uSTAT3) can regulate the expression of *Mras*, *Met*, *Cyclin B1*, and *E2f1* genes (*Srivastava and DiGiovanni, 2016*; *Yang et al., 2005*; *Yang et al., 2007*). Due to the positive roles in cancer development, STAT3 has become an appealing drug target for anti-cancer therapy. So far, a few STAT3 inhibitors have been developed (*Yang et al., 2019a*; *Xiang et al., 2016*; *Bao et al., 2017*; *Shou et al., 2016*; *Rajendran et al., 2011*; *Yang et al., 2013*; *Zhang et al., 2017*). Recently, Wang lab found that the STAT3 inhibitor SD-36 can effectively degrade STAT3 protein and severely suppress the growth of a subset of acute myeloid leukemia by inducing cell arrest and apoptosis (*Bai et al., 2019*). Thus, it is necessary to further investigate and understand the role and regulatory mechanism of STAT3 in cancer cell proliferation and survival to develop effective anti-cancer drugs.

In our recent study, we found that STAT3 can positively regulate RNA polymerase I-dependent transcription by controlling RPA34 expression (manuscript accepted). Since 5S rRNA, an important component of ribosomes, was synthesized by RNA polymerase III, we hypothesized that STAT3 can also regulate Pol III-dependent transcription to coordinate cellular ribosome biogenesis. Indeed, in this study, we found that STAT3 expression levels positively correlate with the synthesis of Pol III products and cell proliferation. Mechanistic analysis revealed that STAT3 modulates Pol III-directed transcription by controlling the activity of the miR-106a-5p-TP73 axis.

## Results

### STAT3 positively regulates Pol III-directed transcription in liver cancer cells

RNA polymerase I and III (Pol I and Pol III) products play fundamental roles in several cellular processes, including ribosomal biogenesis, translation, development, and cell growth (*Watt et al., 2023*). In our recent work, we have shown that STAT3 can enhance Pol I-dependent transcription by controlling RPA34 expression (manuscript accepted). Since 5S rRNA transcribed by Pol III is an important component of ribosomes, we hypothesized that STAT3 might also modulate Pol III-directed transcription to coordinate cellular ribosomal biogenesis. To prove this hypothesis, we initially generated two liver cancer cell lines stably expressing STAT3 short hairpin RNA (shRNA) or control shRNA using a lentiviral transduction system (*Figure 1A and B*, *Figure 1—figure supplement 1A and B*) and assessed the effect of STAT3 depletion on the expression of Pol III products by RT-quantitative PCR (qPCR). Interestingly, STAT3 silencing in HepG2 and HuH7 cells severely reduced the expression of Pol III products, including 5S rRNA, U6 snRNA, and 7 SL RNA (*Figure 1C* and *Figure 1—figure supplement 1C*). To confirm whether this observation can be reproduced in other cell lines, we generated 293T cell lines stably expressing STAT3 shRNA or control shRNA (*Figure 1D and E*) and analyzed the synthesis of Pol III products by RT-qPCR. As expected, RT-qPCR results were consistent with those obtained in the assays with HepG2 or HuH-7 cells (*Figure 1F*). These results suggest that STAT3 is essential to maintaining normal transcription directed by Pol III and functions as a positive regulator in this process. To validate the positive role of STAT3 in Pol III-directed transcription, we generated several cell lines with STAT3 overexpression, including HepG2, HuH-7, and 293T cell lines (*Figure 1G and I*, *Figure 1—figure supplement 1D*). Analysis of Pol III products showed that STAT3 overexpression stimulated the expression of Pol III products, including 5S rRNA, U6 snRNA, and 7SL RNA expression (*Figure 1H and J*, *Figure 1—figure supplement 1E*). tRNA molecules are a major subset of Pol III products; thus, we analyzed the effect of STAT3 silencing and overexpression on the expression of tRNA genes in the cell lines achieved above. The expression of tRNA genes randomly selected for the assay was analyzed by RT-qPCR. Noticeably, STAT3 silencing dampened the expression of tRNA genes detected in the assay (*Figure 1K and L*, *Figure 1—figure supplement 1F*). In contrast, STAT3 overexpression enhanced the expression of the same tRNA genes (*Figure 1M and N*, *Figure 1—figure supplement 1G*). Taken together, these results indicate that STAT3 can positively regulate Pol III-directed transcription in 293T and liver cancer cells.

### Alteration of Pol III products caused by STAT3 silencing or overexpression affected liver cancer cell growth in vitro and in vivo

It is well established that Pol III product levels correlate closely with cell growth or cell proliferation activity (*Moir and Willis, 2013*; *Turowski and Tollervey, 2016*; *Leśniewska and Boguta, 2017*). Whether alteration of Pol III products mediated by STAT3 can affect cell proliferative activity remains to be addressed. To this end, we first analyzed the proliferative activity of HepG2 cell lines with STAT3 silencing or overexpression using cell counting and cell counting kit-8 (CCK-8) methods. For proliferation assays by CCK-8, the water-soluble tetrazolium salt WST-8 (2-[2-methoxy-4-nitrophenyl]–3-[4-n itrophenyl]–5-[2,4-disulfophenyl]–2H-tetrazolium, monosodium salt) was added cell culture medium, where the compound was reduced by cellular dehydrogenase and became a water-soluble yellow dye (formazan) easily monitored by a spectrometer. Interestingly, STAT3 silencing reduced the proliferative activity of HepG2 cells (*Figure 2A and B*). In contrast, STAT3 overexpression promoted HepG2 cell proliferation (*Figure 2C and D*). Consistent results were obtained when cell proliferation assays were performed using HuH-7 and 293T cell lines with STAT3 depletion or overexpression (*Figure 2—figure supplement 1*). A deoxyuridine analog, EdU (5-ethynyl-2'-deoxyuridine), can be easily incorporated into the newly synthesized DNA and widely used for cell proliferation assays. Thus, we determined the effect of STAT3 expression alteration on the efficacy of EdU labeling in HepG2. As expected, STAT3 depletion blunted the rate of EdU-labeled cells (*Figure 2E and F*); whereas STAT3 upregulation enhanced the rate of EdU-labeled cells, indicating that STAT3 can promote the proliferative activity of liver cancer cells.

STAT3 has been shown to regulate both cell proliferation and Pol III-directed transcription. Thus, we next investigated whether changes in cell proliferation are caused by the alteration of Pol III

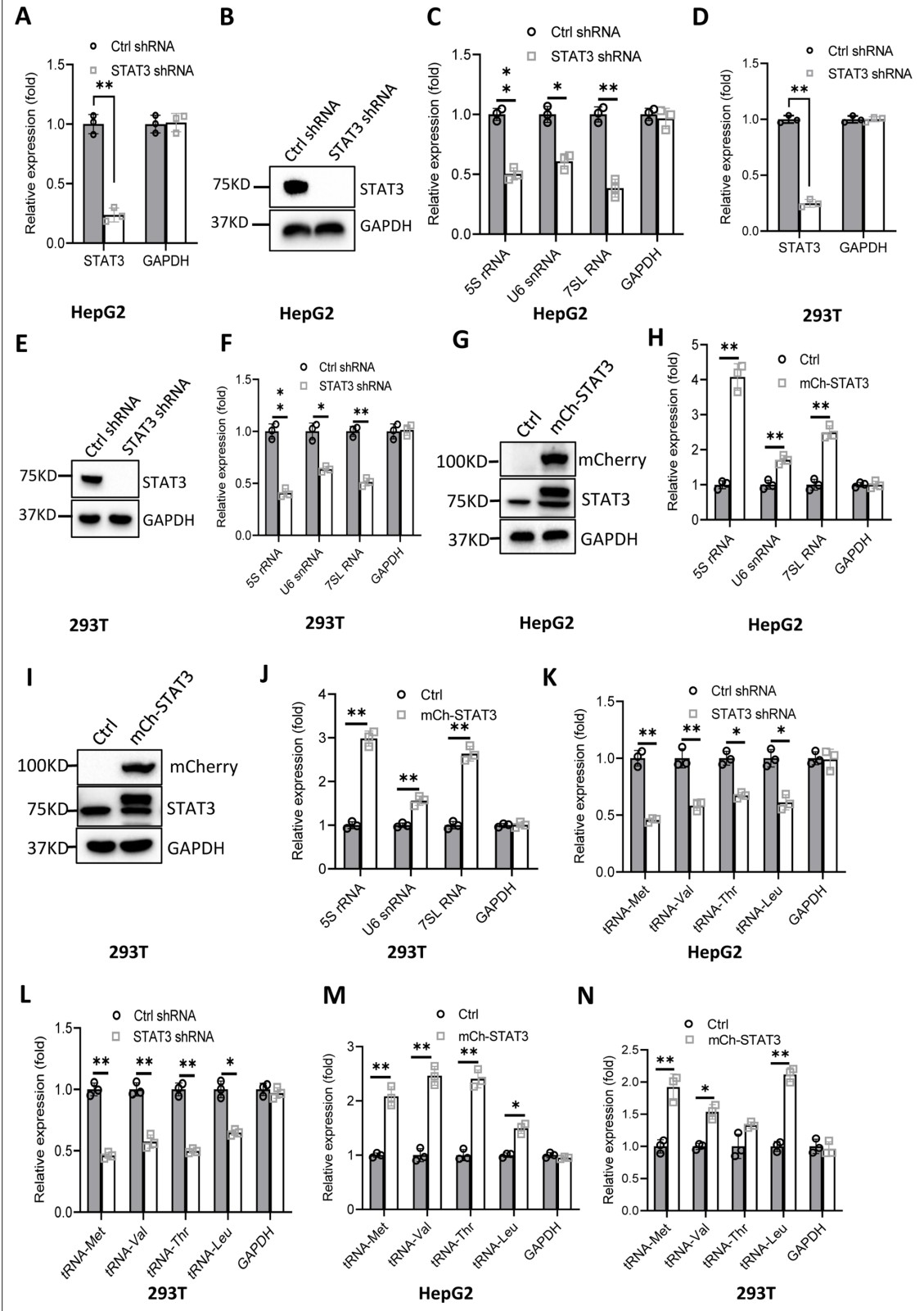

**Figure 1.** Effect of STAT3 expression alteration on the synthesis of Pol III products. (**A–C**) STAT3 knockdown reduced the synthesis of Pol III products in HepG2 cells. HepG2 cell lines stably expressing STAT3 shRNA or control shRNA were generated by a lentiviral transduction system. STAT3 expression was analyzed by RT-quantitative PCR (qPCR) (**A**) and western blot (**B**). Pol III products were monitored by RT-qPCR (**C**). (**D–F**) STAT3 knockdown decreased the synthesis of Pol III products in 293T cells. 293T cell lines stably expressing STAT3 shRNA or control shRNA were generated as described

*Figure 1 continued on next page*

*Figure 1 continued*

in A and B. STAT3 expression (**D and E**) and Pol III products (**F**) were detected as described in A–C. (**G and H**) STAT3 overexpression activated the expression of Pol III products in HepG2 cells. A HepG2 cell line stably expressing mCherry-STAT3 shRNA and its control cell line was generated by a lentiviral transduction system. STAT3 protein and Pol III products were analyzed by western blot (**G**) and RT-qPCR (**H**), respectively. (**I and J**) STAT3 overexpression enhanced the expression of Pol III products in 293 cells. A 293T cell line stably expressing mCherry-STAT3 shRNA and its control cell line were generated using a lentiviral transduction system. STAT3 protein and Pol III products were detected by western blot (**I**) and RT-qPCR (**J**), respectively. (**K and L**) STAT3 silencing inhibited the expression of tRNA genes. The expression of tRNA genes randomly selected was monitored by RT-qPCR using HepG2 (**K**) and 293T (**L**) cell lines with STAT3 depletion. (**M and N**) STAT3 overexpression activated the expression of tRNA genes. HepG2 (**M**) and 293T (**N**) cell lines with STAT3 overexpression were used to analyze the expression of tRNA genes by RT-qPCR. Each column in A, C, D, F, H, and J–N represents the mean ± SD of three biological replicates (n=3). *, p<0.05; **, p<0.01. p Values were obtained by Student's *t* test performed with data for the control and treated samples. Mean, SD and p values were calculated using GraphPad Prism 8 software.

The online version of this article includes the following source data and figure supplement(s) for figure 1:

**Source data 1.** Raw images for Western blot data in *Figure 1B, E, G and I*.

**Source data 2.** Original digital data in *Figure 1*.

**Figure supplement 1.** Effect of STAT3 expression alteration on Pol III-dependent transcription in HuH-7 cells.

**Figure supplement 1—source data 1.** Raw images for Western blot data in *Figure 1—figure supplement 1*.

**Figure supplement 1—source data 2.** Original digital data for *Figure 1—figure supplement 1*.

products. To this end, cell proliferation assays were performed using the HepG2 cell lines with or without STAT3 overexpression in the presence and absence of Pol III transcription-specific inhibitor (ML-60218, 54 µM). We showed that STAT3 overexpression enhanced cell proliferation and Pol III-dependent transcription and that the presence of ML-60218 inhibited proliferative activity and pol III-dependent transcription for HepG2 control cells. Strikingly, the presence of ML-60218 reversed the activation of cell proliferation and Pol III-directed transcription induced by STAT3 overexpression (*Figure 2—figure supplement 2A and B*), indicating that Pol III transcription activation contributed to the enhancement of cell proliferation induced by STAT3 overexpression, although the contribution of other factors cannot be excluded.

To validate whether STAT3 promotes liver cancer cell growth in vivo, we injected HepG2 cells into nude mice and measured tumor sizes regularly during tumor formation. Analysis of tumor sizes showed that STAT3 silencing inhibited the sizes of liver tumors during tumor formation (*Figure 2I*). After removing from nude mice, liver tumors were photographed, weighed, and subjected to statistical analysis. Interestingly, STAT3 silencing reduced the weights of tumors compared to the weight of tumors derived from HepG2 control cells. Next, tumor tissues randomly selected were used for immunohistochemical staining. Noticeably, tumor tissues derived from STAT3-depleted HepG2 cells exhibited lower expression of STAT3 compared to those derived from HepG2 control cells. These results indicate that STAT3 silencing suppresses liver cancer cell growth in vivo. STAT3 has been shown to modulate cell survival and proliferation (*Srivastava and DiGiovanni, 2016*; *Johnson et al., 2018*), whether alteration of liver cancer cell growth mediated STAT3 upregulation or downregulation is caused by both cell survival and proliferation remains unclear. To address this question, we performed colony formation assays using HepG2 cell lines with STAT3 silencing or overexpression. Colony formation assays showed that STAT3 depletion reduced the number of colonies and the size of individual colonies (*Figure 2—figure supplement 3A–C*). In contrast, STAT3 overexpression increased the number of colonies and the size of individual colonies significantly (*Figure 2—figure supplement 3D–F*), indicating that both cell survival and proliferative activity contribute to the alteration of cell growth induced by STAT3 silencing or overexpression.

## STAT3 silencing decreased the recruitment of Pol III transcription machinery components to Pol III target loci

To understand how STAT3 regulates Pol III-dependent transcription, we performed chromatin immunoprecipitation (ChIP) assays using HepG2 cell lines stably expressing STAT3 shRNA or control shRNA and antibodies against components of the Pol III transcription machinery. The results of ChIP assays showed that STAT3 knockdown reduced the recruitment of TBP, BRF1, GTF3C2, GTF3C3, and POLR3A at several Pol III target loci, including DNA loci encoding 5S rRNA, 7SL RNA, and tRNA-Met (*Figure 3A–H*). However, the occupancy of BRF1, GTF3C2, and GTF3C3 at the DNA locus encoding

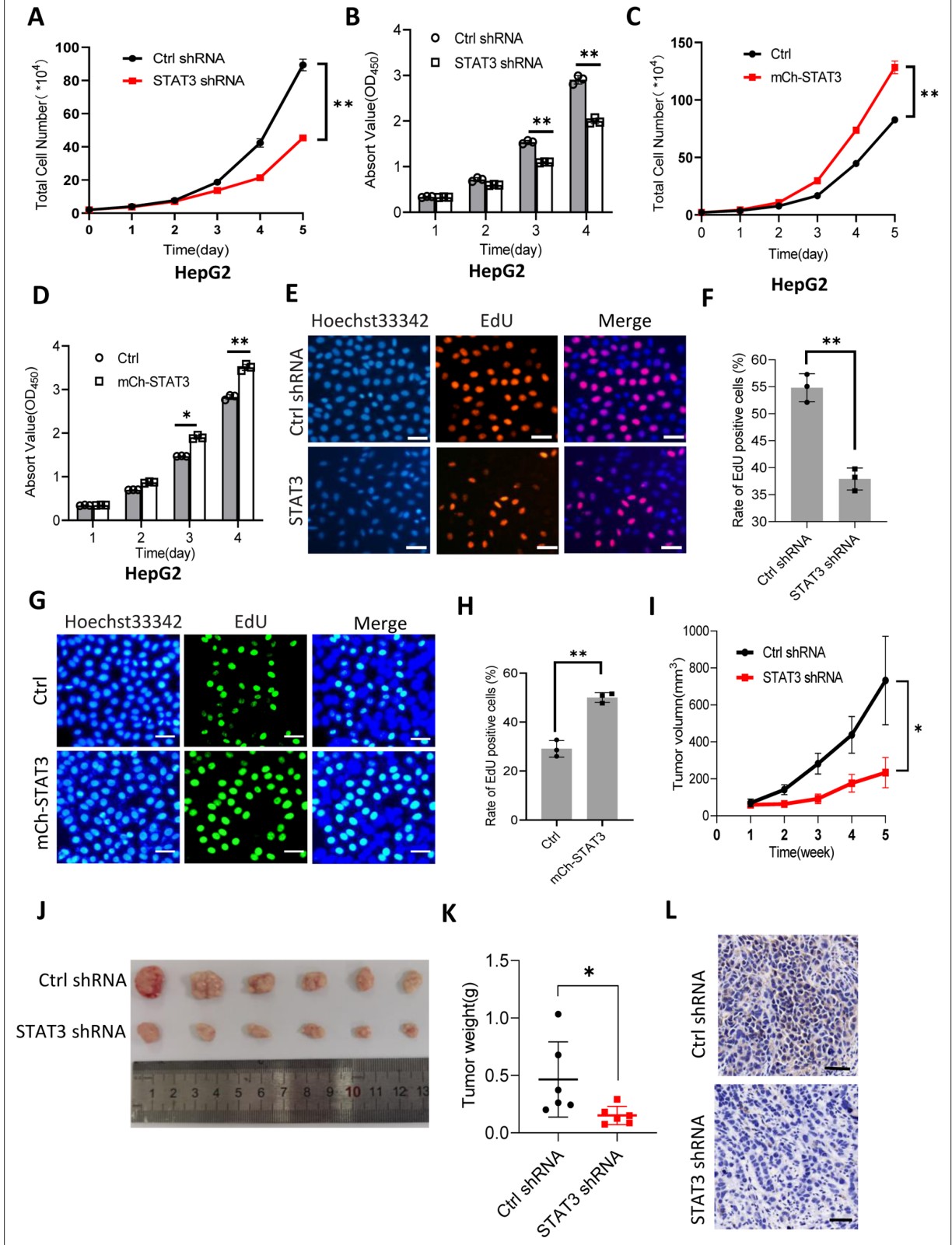

**Figure 2.** STAT3 promotes cell growth in vitro and in vivo. (**A and B**) STAT3 silencing reduced the activity of HepG2 cell proliferation. Proliferative activity was determined by cell counting (**A**) and CCK-8 assays (**B**). (**C and D**) STAT3 overexpression enhanced the activity of HepG2 cell proliferation. Proliferative activity was determined by cell counting (**C**) and CCK-8 assays (**D**). (**E and F**) EdU assay results for HepG2 cells line expressing STAT3 shRNA or control shRNA. EdU-labeled cells were imaged under a fluorescent microscope (**E**), and the rate of EdU positive cells (**F**) was analyzed by

*Figure 2 continued*

ImageJ software based on the images obtained in E. Scale bars in C represent 50 µm. (**G and H**) EdU assay results for the HepG2 cell line with STAT3 overexpression and its control cell line. EdU-labeled cells were imaged under a fluorescent microscope (**G**) and the rate of EdU positive cells (**H**) was analyzed by ImageJ software based on the images obtained in G. Scale bars in C represent 50 µm. (**I**) The time course of tumor volumes during tumor formation. Amount of $1 \times 10^7$ HepG2 cells expressing Ctrl shRNA or STAT3 shRNA were injected subcutaneously into nude mice (n=6). After 1 week, tumor sizes were measured by vernier caliper. (**J and K**) STAT3 silencing reduced the weights of tumors formed in nude mice. After nude mice were euthanized, tumors were taken out of the mice and pictured with a camera (**J**); the resulting tumors were weighed, and the resulting data were analyzed statistically (**K**). (**L**) STAT3 immunohistochemical staining for the tumor samples derived from HePG2 cells expressing control shRNA or STAT3 shRNA. Scale bars in the images represent 100 µm. Each column or point in histograms represents the mean ± SD of three biological replicates (n=3). *, p<0.05; **, p<0.01. p Values were obtained by Student's *t* test (B, D, F, H, and K) or two-way ANOVA (A, C, and I). Mean, SD, and p values were calculated using GraphPad Prism 8 software.

The online version of this article includes the following source data and figure supplement(s) for figure 2:

**Source data 1.** Original digital data for *Figure 2*.

**Figure supplement 1.** Effect of STAT3 expression alteration on the activity of cell proliferation.

**Figure supplement 1—source data 1.** Original digital data for *Figure 2—figure supplement 1*.

**Figure supplement 2.** Effect of ML-60218 on cell proliferation and Pol III product expression in HepG2 cell lines with STAT3 overexpression.

**Figure supplement 2—source data 1.** Original digital data for *Figure 2—figure supplement 2*.

**Figure supplement 3.** Alteration of STAT3 expression affected colony formation for HepG2 cells.

**Figure supplement 3—source data 1.** Original digital data for *Figure 2—figure supplement 3*.

U6 snRNA showed a comparable level to that of Control IgG (*Figure 3C and D*). This result is reasonable because transcription of U6 snRNA does not require the involvement of BRF1 and TFIIIC (*Moir and Willis, 2013*; *Turowski and Tollervey, 2016*). Interestingly, the occupancy of TBP, BRF2, and POLR3A was reduced at the U6 snRNA promoter after STAT3 depletion. To obtain the details about the mechanism by which STAT3 regulates the assembly of Pol III transcription machinery at its target loci, we determined whether STAT3 binds to Pol III target loci by performing ChIP assays using HepG2 cells. Unexpectedly, STAT3 does not bind to the DNA loci that encode 5S rRNA, U6 snRNA, 7SL RNA, and tRNA-Met, but it binds to the promoter of STAT3 target gene *BCL2* (*Figure 3I*), suggesting that STAT3 does not directly regulate Pol III-directed transcription. We next determined whether STAT3 depletion affects the expression of Pol III transcription machinery components. Expression of Pol III transcription factors was analyzed by western blot assays using HepG2 and 293T cell lines expressing STAT3 shRNA or control shRNA and antibodies against TBP, BRF1, GTF3C2, and GTF3C3. Western blot results showed that STAT3 silencing did not affect the expression of Pol III transcription machinery components in HepG2 and 293T cells (*Figure 3J and K*), suggesting that the occupancy changes of Pol III transcription machinery components at Pol III target loci are not caused by the expression of Pol III general transcription factor subunits tested in the assays.

## STAT3 inhibits TP73 expression

We showed that STAT3 knockdown did not affect the expression of TBP, BRF1, GTF3C2, and GTF3C3 proteins. However, the number of Pol III transcription machinery components is far beyond the number of factors tested above. Thus, we sought to investigate the expression of all components of the Pol III transcription machinery by performing RNA-seq using the STAT3-depleted HepG2 cells and the corresponding control cells. Analysis of mRNA transcriptome showed that STAT3 silencing caused expression upregulation of 356 genes and expression downregulation of 590 genes significantly (*Figure 4A and B*). Next, we analyzed the expression of genes encoding components of the Pol III transcription machinery by searching the RNA-seq data. The results showed that STAT3 knockdown did not alter the mRNA expression of components of the Pol III transcription machinery significantly (*Figure 4C*), suggesting that STAT3 regulates Pol III-dependent transcription through alternative pathways. Alteration of STAT3 expression can affect the synthesis of Pol III products and the proliferative activity of several cell lines (*Figures 1 and 2*). Furthermore, Pol III product levels correlate closely with cell proliferative activity (*Willis and Moir, 2018*; *Yeganeh and Hernandez, 2020*). Therefore, we analyzed the differential expression genes (DEGs) that are relevant with cell proliferation. Interestingly, dozens of the DEGs were identified from several proliferation-related pathways based on the RNA-seq data (*Figure 4D*, *Figure 4—figure supplement 1A*). Among these genes, TP73, a tumor suppressor, is

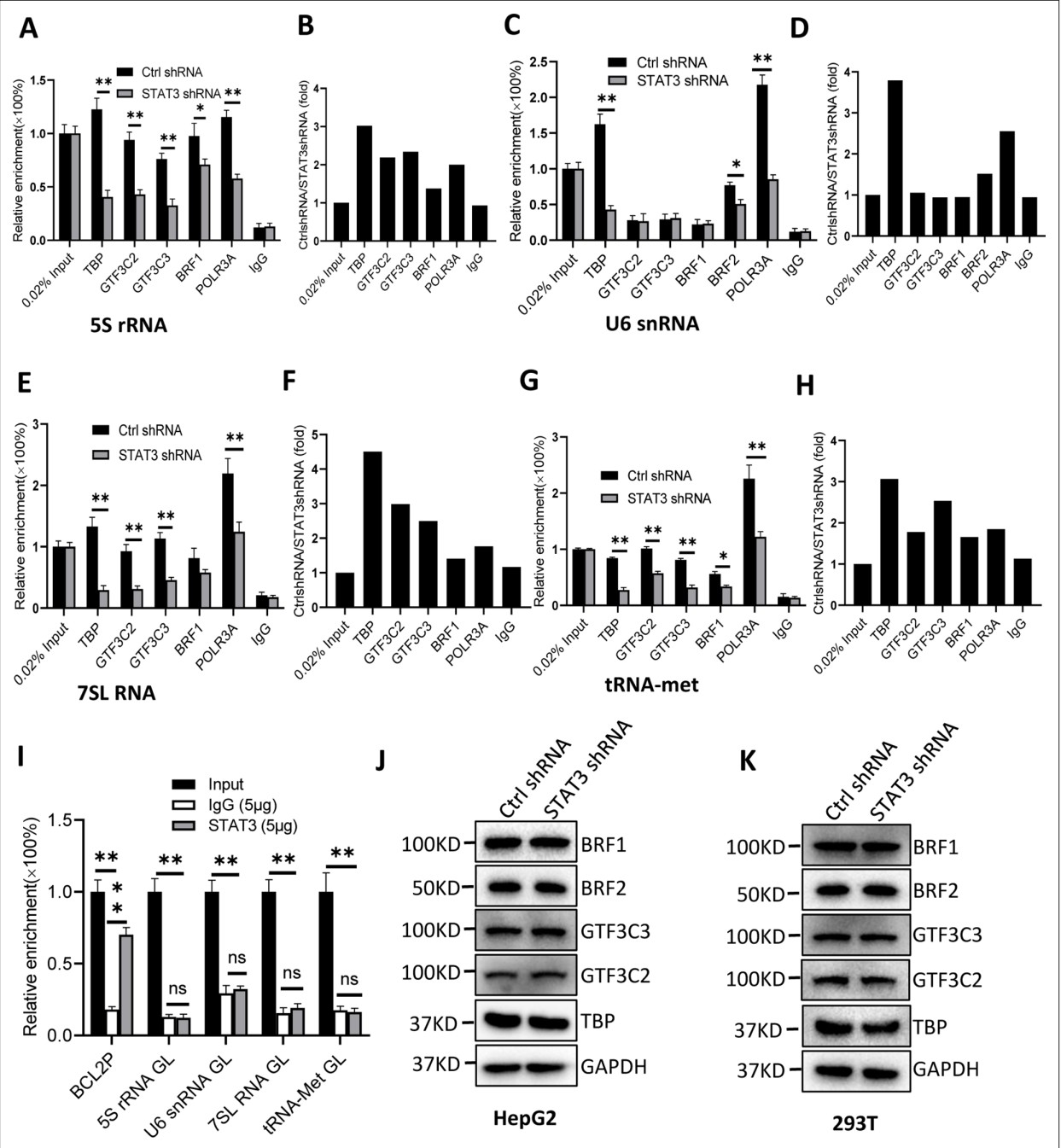

**Figure 3.** STAT3 silencing inhibited the recruitment of Pol III transcription machinery components to Pol III target loci. (**A and B**) STAT3 knockdown reduced the occupancies of Pol III transcription machinery components at the DNA locus encoding 5S rRNA. Chromatin immunoprecipitation (ChIP) assays were performed using HepG2 cell lines stably expressing Ctrl shRNA or STAT3 shRNA and antibodies against the indicated factors. Relative enrichment (**A**) was obtained by comparing the quantity of target DNA in 1 μL of ChIP DNA sample to that in 0.02% of input DNA sample. Fold change (**B**) was obtained by comparing the mean of relative enrichment of target DNA in a control shRNA sample to that in a STAT3 shRNA sample obtained in A. (**C and D**) STAT3 knockdown decreased the recruitment of TBP, BRF2, and POLR3A to the DNA locus encoding U6 snRNA. Relative enrichment (**C**) and fold change (**D**) were obtained as described in A and B, respectively. (**E and F**) STAT3 knockdown inhibited the recruitment of Pol III transcription machinery components to the DNA locus encoding 7SL RNA. Relative enrichment (**E**) and fold change (**F**) were obtained as described in A and B, respectively. (**G and H**) STAT3 knockdown dampened the recruitment of Pol III transcription machinery components to the tRNA-Met promoter. Relative enrichment (**G**) and fold change (**H**) were obtained as described in A and B, respectively. (**I**) STAT3 did not bind to the DNA loci encoding 5S rRNA, U6 RNA, 7SL RNA, and tRNA-met but *BCL2* promoter. ChIP assays were performed using HepG2 cells and an anti-STAT3 antibody. Relative enrichment (**G**) was obtained as described in A. (**J and K**) STAT3 silencing did not affect the expression of Pol III transcription factor subunits. HepG2 (**J**) and 293T (**K**) cell lines with STAT3 depletion were used to analyze the expression of pol III transcription factor subunits by western blot. Each column in A, C, E,

*Figure 3 continued on next page*

*Figure 3 continued*

G, and I represents the mean of SD of three biological replicates (n=3) *, p<0.05; **, p<0.01. p Values were obtained by Student's *t* test performed with data for the control and treated samples. Mean, SD, and p values were calculated using GraphPad Prism 8 software.

The online version of this article includes the following source data for figure 3:

**Source data 1.** Raw images for Western blot data in *Figure 3*.

**Source data 2.** Original digital data for *Figure 3*.

a member of the p53 family (*Maeso-Alonso et al., 2021*) and exhibited increased expression after STAT3 silencing (*Figure 4D*). Both DPT and BMP2 can play roles in the TGFβ signaling pathway (*Okamoto et al., 1999*; *Chen et al., 2021*), and their expression was increased or decreased by STAT3 depletion (*Figure 4D*). FGFR3 activating RAS-ERK or PTEN/AKT signal pathways (*Daly et al., 2017*; *Jin et al., 2020*) showed reduced expression after STAT3 downregulation (*Figure 4—figure supplement 1B*). TGFβ has been shown to regulate the activity of several signal pathways, including ERK/MAP, PTEN/AKT, and JNK pathways (*Yang et al., 2021*); these pathways together with p53 signaling have been reported to modulate Pol III-directed transcription (*Moir and Willis, 2013*; *Willis and Moir, 2018*; *Crighton et al., 2003*). Therefore, we investigated the effect of STAT3 expression alteration on the expression of DPT, TP73, BMP2, and FGFR3 in HepG2 cells. RT-qPCR results showed that STAT3 silencing stimulated TP73 expression but inhibited BMP2 and FGFR3 expression. DPT expression remained unchangeable. In contrast, STAT3 overexpression inhibited TP73 expression, but the expression of other factors was not affected by STAT3 overexpression significantly (*Figure 4E and F*). Analysis of STAT3 and TP73 reads revealed that STAT3 silencing increased TP73 mRNA reads (*Figure 4G and H*). Interestingly, STAT3 expression in SaOS2 cells is also opposite to TP73 expression (*Figure 4—figure supplement 2*) when gene expression is analyzed based on the RNA-seq data derived from SaOS2 FLNA-depleted cells (*Zhang et al., 2022*). Collectively, these results suggest that STAT3 can negatively regulate TP73 mRNA expression.

## TP73 inhibits Pol III-directed transcription and cell proliferation

STAT3 has been shown to inhibit TP73 mRNA expression; thus, we determined whether TP73 protein expression is affected by STAT3 expression changes. Western blot analysis showed that STAT3 depletion enhanced TP73 protein expression, whereas STAT3 overexpression reduced its protein levels in HepG2 cells (*Figure 5A–D*). Consistent results were obtained when TP73 expression was analyzed using HuH-7 and 293T cells with STAT3 silencing or overexpression (*Figure 5—figure supplement 1*). These results together with the findings in *Figure 4* indicate that STAT3 can suppress TP73 expression at both mRNA and protein levels. To understand whether TP73 is required for the regulation of Pol III-directed transcription and cell proliferation mediated by STAT3, we generated HepG2 cell lines stably expressing both STAT3 shRNA and TP73 shRNA based on HepG2 cells with STAT3 silencing (*Figure 5E*). Analysis of Pol III products showed that TP73 shRNA stable expression reversed the inhibition of Pol III product levels caused by STAT3 silencing (*Figure 5F*). Cell proliferation assays revealed that TP73 knockdown also reversed the suppression of proliferative activity caused by STAT3 silencing (*Figure 5G*), indicating that TP73 plays a negative role in the regulation of Pol-III-directed transcription and cell proliferation mediated by STAT3. We next determined whether TP73 can independently regulate Pol III-dependent transcription and proliferative activity. To achieve this goal, we generated HepG2 cell lines expressing TP73 shRNA or control shRNA (*Figure 5H and I*) and analyzed the expression of Pol III products by RT-qPCR using the cell lines established. Interestingly, TP73 knockdown enhanced Pol III product levels and HepG2 cell proliferation activity (*Figure 5J–L*), suggesting that TP73 inhibits Pol III-dependent transcription and cell proliferation. To verify this result, we transfected HepG2 cells with the plasmids expressing dCAS9-VP48 and gRNA to target the TP73 promoter and observed the effect of endogenous TP73 expression activation on Pol III products and cell growth. As expected, endogenous TP73 expression activation caused the inhibition of Pol III products and cell proliferation (*Figure 5—figure supplement 2*), indicating TP73 can independently inhibit Pol III-dependent transcription. To understand whether the increase of Pol III products contributes to the enhancement of cell proliferation induced by TP73 silencing, we analyzed Pol III products and proliferative activity using HepG2 cells expressing TP73 shRNA or control shRNA in the presence or absence of ML-60218. As expected, ML-60218 inhibited the activation of Pol III-directed transcription

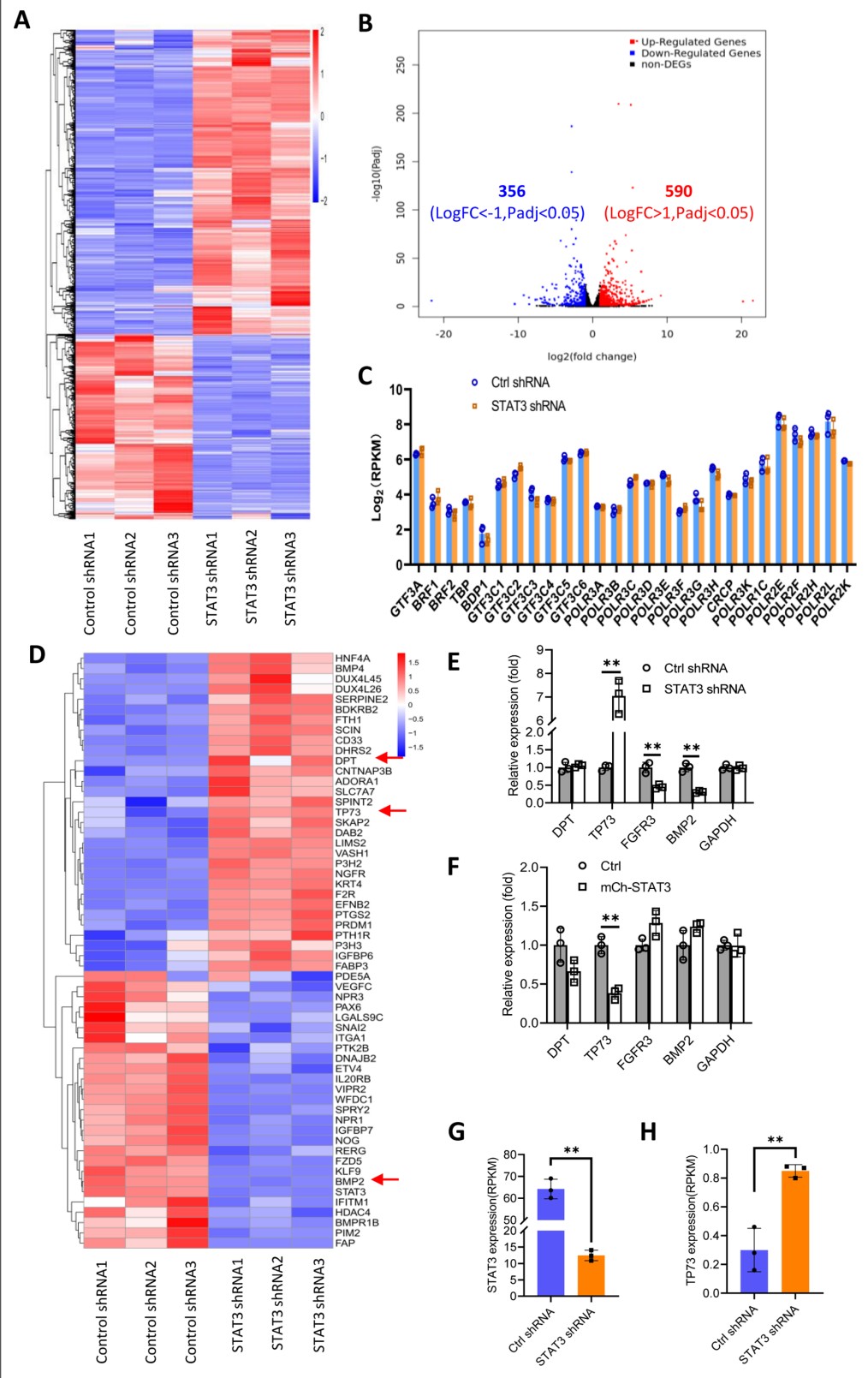

**Figure 4.** Effect of STAT3 silencing on genome-wide mRNA expression in HePG2 cells. (**A**) A Heatmap showing the result of hierarchical cluster analysis based on the mRNA-seq data derive from HepG2 cell lines stably expressing control shRNA or STAT3 shRNA. (**B**) A volcano plot showing the number of significantly upregulated or downregulated genes obtained from the analysis of the RNA-seq data. (**C**) STAT3 silencing did not affect the

*Figure 4 continued on next page*

*Figure 4 continued*

expression of genes encoding Pol III transcription machinery components. Gene expression (log$_2$[RPKM]) was obtained from the RNA-seq data. (**D**) A heatmap showing differential expression genes related to the pathways for negative regulation of cell proliferation. Expression for *DPT*, *TP73*, and *BMP2* genes was pointed out by red arrows. (**E and F**) Effect of STAT3 silencing and overexpression on the expression of *DPT*, *TP73*, *FGFR3*, and *BMP2* genes. RT-quantitative PCR (qPCR) was performed using HepG2 cell lines with STAT3 silencing (**E**) or overexpression (**F**). (**G and H**) The mRNA reads (RPKM) of *STAT3* and *TP73* genes obtained from the RNA-seq with HepG2 cells expressing STA3 shRNA or control shRNA. Each column in C and E–H represents the mean of SD of three biological replicates (n=3). **, p<0.01. p Values were obtained by Student's *t* test performed with data for the control and treated samples. Mean, SD, and p values were calculated using GraphPad Prism 8 software.

The online version of this article includes the following source data and figure supplement(s) for figure 4:

**Source data 1.** Original digital data for *Figure 4*.

**Figure supplement 1.** Effect of STAT3 silencing on the expression of genes related to cell proliferation.

**Figure supplement 1—source data 1.** Original digital data for *Figure 4—figure supplement 1*.

**Figure supplement 2.** TP73 expression is opposite to STAT3 in FLNA-depleted cells.

**Figure supplement 2—source data 1.** Original digital data for *Figure 4—figure supplement 2*.

---

and proliferative activity induced TP73 silencing (*Figure 5—figure supplement 3*), indicating that changes in Pol III product levels are associated with the promotion of cell proliferation induced by TP73 silencing. This result is in agreement with that obtained in *Figure 2—figure supplement 2*; further confirming that alteration of Pol III products can affect cell proliferative activity. Taken together, TP73 is not only required for the regulation of Pol III-directed transcription mediated by STAT3 but also independently modulates Pol III-dependent transcription.

## TP73 inhibits Pol III-directed transcription by disrupting TFIIIB assembly

TP73 has been shown to negatively regulate Pol III-directed transcription. To uncover the mechanism by which TP73 modulates Pol III-directed transcription, we performed ChIP assays using HepG2 cell lines with TP73 depletion and antibodies against components of the Pol III transcription machinery. Interestingly, TP73 silencing enhanced the occupancies of Pol III transcription machinery components at DNA loci encoding 5S rRNA, 7SL RNA, and tRNA-Met (*Figure 6A, B and E–H*). For the DNA locus encoding U6 snRNA, however, TBP, BRF2, and POLR3A exhibited increased occupancy at this locus after TP73 depletion, but the occupancies of BRF1, GTF3C2, and GTF3C3 showed similar levels to that of Ctrl IgG (*Figure 6C and D*). These results suggest that TP73 suppresses Pol III-dependent transcription by reducing the recruitment of Pol III transcription machinery components to Pol III target loci. To understand how TP73 regulates the assembly of Pol III transcription machinery components at Pol III-transcribed loci, we analyzed the effect of TP73 silencing on the expression of Pol III transcription machinery components by western blot. Unexpectedly, TP73 depletion did not affect the expression of Pol III transcription machinery components, including TBP, BRF1, BRF2, GTF3C2, and GTF3C3 (*Figure 6I*), indicating that TP73 regulates the Pol III transcription machinery assembly at Pol III target loci via other mechanisms. To deepen the understanding of the mechanism by which TP73 regulates Pol III-directed transcription, we performed co-immunoprecipitation (co-IP) assays using an anti-TP73 antibody. Immunoprecipitation (IP) results showed that TBP, GTF3C2, and GTF3C3 were precipitated by the anti-TP73 antibody, but BRF1 was not precipitated by the anti-TP73 antibody (data not shown). In reciprocal assays, TP73 was precipitated by TBP, GTF3C2, and GTF3C3 (*Figure 6J–L*). These results indicate that TP73 can interact with TBP, GTF3C2, and GTF3C3. We next addressed how TP73 inhibits Pol III-mediated transcription by interacting with Pol III general transcription factors. Co-IP assays were performed using HepG2 cells expressing TP73 shRNA or control shRNA and antibodies against TBP, BRF1, and BRF2. Strikingly, TP73 silencing enhanced TBP binding to BRF1 when an anti-TBP antibody was used for co-IP assays (*Figure 6M*). In reciprocal IP assays, a consistent result was obtained (*Figure 6N*). Furthermore, this observation has also been confirmed by the co-IP assays using the antibodies against TBP and BRF2 (*Figure 6—figure supplement 1*). These results suggest that TP73 can interfere with TFIIIB assembly by interacting with TFIIIB subunit TBP. Collectively, TP73 can inhibit Pol III-directed transcription by affecting TFIIIB assembly.

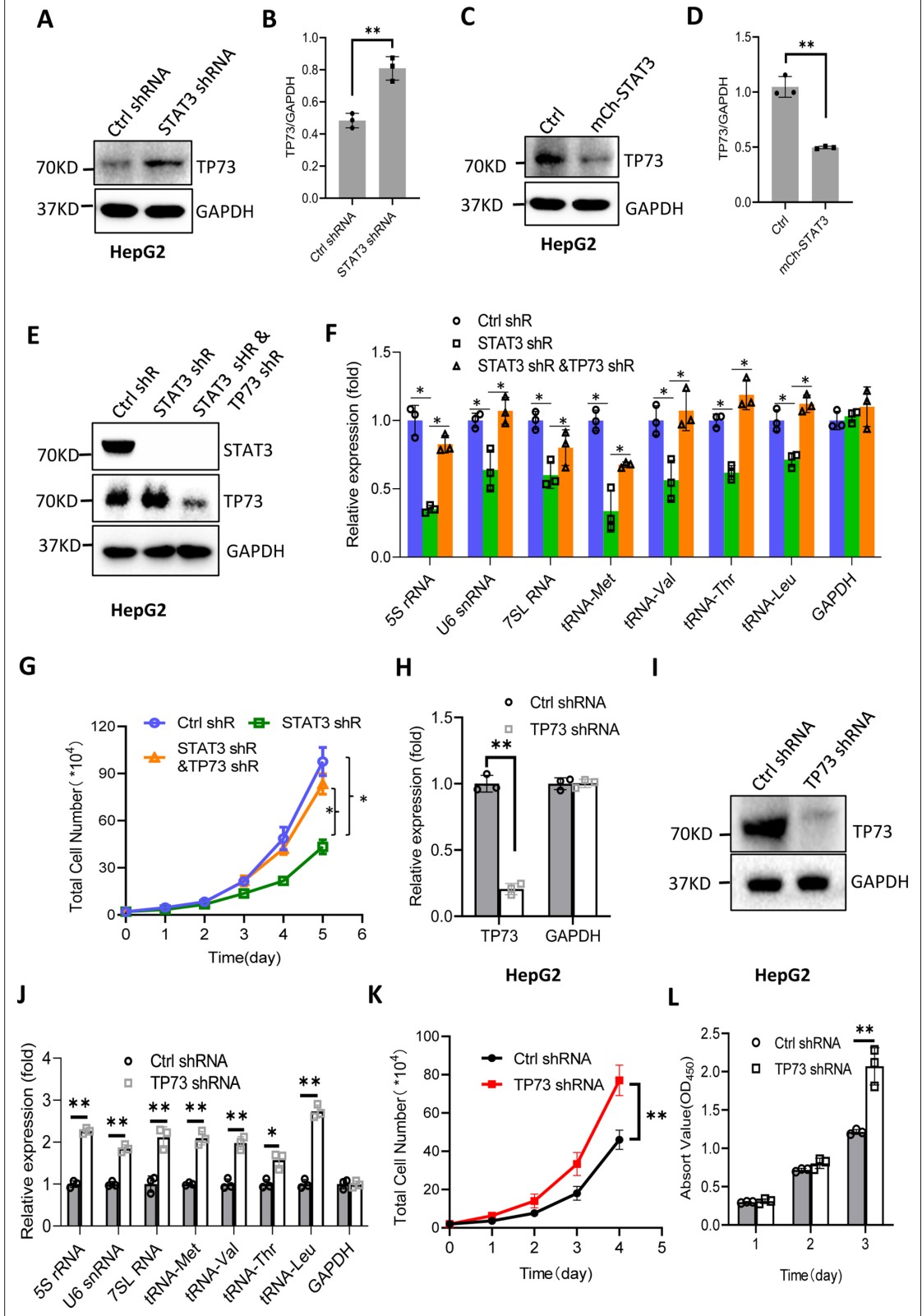

**Figure 5.** STAT3 modulates Pol III-directed transcription by inhibiting TP73 expression. (**A and B**) STAT3 knockdown stimulated TP73 protein expression. HepG2 cells expressing STAT3 shRNA or control shRNA were cultured and harvested for western blot analysis. B is the quantified result for western blots obtained in A. (**C and D**) STAT3 overexpression reduced TP73 protein expression. Western blot was performed using the HepG2 cell line with STAT3 overexpression and the corresponding control cell line. D is the quantified result for western blots obtained in C. (**E**) Western blot results showing

*Figure 5 continued on next page*

*Figure 5 continued*

the generation of HepG2 cell lines expressing both STAT3 shRNA and TP73 shRNA. shR: shRNA. (**F**) TP73 silencing reversed the inhibition of Pol III product levels induced by STAT3 depletion. RT-quantitative PCR (qPCR) was performed using the cell lines generated in E. (**G**) TP73 silencing reversed the inhibition of cell proliferation caused by STAT3 depletion. Cell counting was performed using the cell lines established in E. (**H and I**) HepG2 cell lines stably expressing TP73 shRNA or control shRNA were generated and verified by RT-qPCR (**H**) and western blot (**I**). (**J**) TP73 silencing enhanced Pol III-directed transcription. RT-qPCR was performed using the cell lines generated in H and I. (**K and L**) TP73 silencing promoted proliferative activity of HePG2 cells. HepG2 cell lines obtained in H and I were used to analyze proliferative activity by cell counting and CCK-8 methods. Each column or point in histograms represents the mean of SD of three biological replicates (n=3). *, p<0.05; **, <0.01. p Values in B, D, F, H, and J were obtained by Student's *t* test performed with the data from control and treated samples; whereas p values in G, K, and L were obtained by two-way ANOVA. Mean, SD, and p values were calculated using GraphPad Prism 8 software.

The online version of this article includes the following source data and figure supplement(s) for figure 5:

**Source data 1.** Raw images for Western blot data in *Figure 5*.

**Source data 2.** Original digital data for *Figure 5*.

**Figure supplement 1.** STAT3 inhibits TP73 expression.

**Figure supplement 1—source data 1.** Raw images for Western blot data in *Figure 5—figure supplement 1*.

**Figure supplement 1—source data 2.** Original digital data for *Figure 5—figure supplement 1*.

**Figure supplement 2.** Activation of endogenous TP72 expression inhibited the synthesis of Pol III products and cell proliferation.

**Figure supplement 2—source data 1.** Raw images for Western blot data in *Figure 5—figure supplement 2*.

**Figure supplement 2—source data 2.** Original digital data for *Figure 5—figure supplement 2*.

**Figure supplement 3.** ML-60218 reversed the activation of Pol III-directed transcription and cell proliferation induced by TP73 silencing.

**Figure supplement 3—source data 1.** Original digital data for *Figure 5—figure supplement 3*.

## miRNA-106a-5p potentially targeting TP73 mRNA is positively regulated by STAT3

We have demonstrated that TP73 regulates Pol III-directed transcription by affecting TFIIIB assembly; however, how STAT3 regulates TP73 remains unclear. STAT3 was originally identified to be a positive regulator in RNA polymerase II-directed transcription (*Srivastava and DiGiovanni, 2016*; *Johnson et al., 2018*; *Lee et al., 2019*), whereas STAT3 was found to inhibit TP73 expression in this study. Therefore, we hypothesized that STAT3 could regulate TP73 expression through affecting the activity of miRNA pathways rather than directly regulating *TP73* gene transcription. To support this hypothesis, we searched for potential miRNA molecules targeting TP73 mRNA from the Database of TargetScan-Human: Prediction of miRNA Targets (*Figure 7—figure supplement 1A*). Intriguingly, many miRNA molecules that are potential to target TP73 mRNA were found. Therefore, we analyzed eight miRNAs with the top scores, and these miRNA molecules were found to derive from four miRNA clusters. We showed that each cluster contains two miRNA molecules, and the distance between two miRNAs within a cluster is adjacent in the genome (*Figure 7—source data 1*). Therefore, we analyzed the expression of four miRNA molecules randomly selected from each miRNA cluster in HepG2 cell lines using RT-qPCR and miRNA detection primers (*Figure 7—figure supplement 1B*). RT-qPCR results showed that STAT3 silencing severely reduced the expression of miR-106a-5p (or miR-106a-5p) but did not significantly affect the expression of other miRNA molecules, including miR-93–5p, miR-519d-3p, and miR-20a-5p expression (*Figure 7A*). Conversely, STAT3 overexpression enhanced expression of miR-106a-5p but did not affect the expression of other miRNAs significantly (*Figure 7B*). To verify this result, we detected miR-106a-5p expression in HuH-7 and 293T cells with STAT3 silencing or overexpression by RT-qPCR, and the results were consistent with those obtained in the assays with HepG2 cells (*Figure 7C–F*). These results indicate that STAT3 can positively regulate miR-106a-5p expression. Next, we investigated whether there is any difference for miR-106a-5p expression between transformed cell lines and their corresponding normal cell lines. RT-qPCR data showed that HepG2 and HEK293T cell lines showed the increase expression of miR-106a-5p compared to the corresponding cell lines HL-7702 and HEK293 (*Figure 7G and H*), suggesting that miR-1065a-5p expression levels have potential biomedical significance. To determine whether the expression levels of miR-106a-5p are associated with the survival probability and time of cancer patients, we performed Kaplan-Meier plotting using the cancer database deposited at The Cancer Genome Atlas (TCGA). Interestingly, liver hepatocellular carcinoma (LIHC) patients with high expression levels of miR-106a-5p exhibit lower

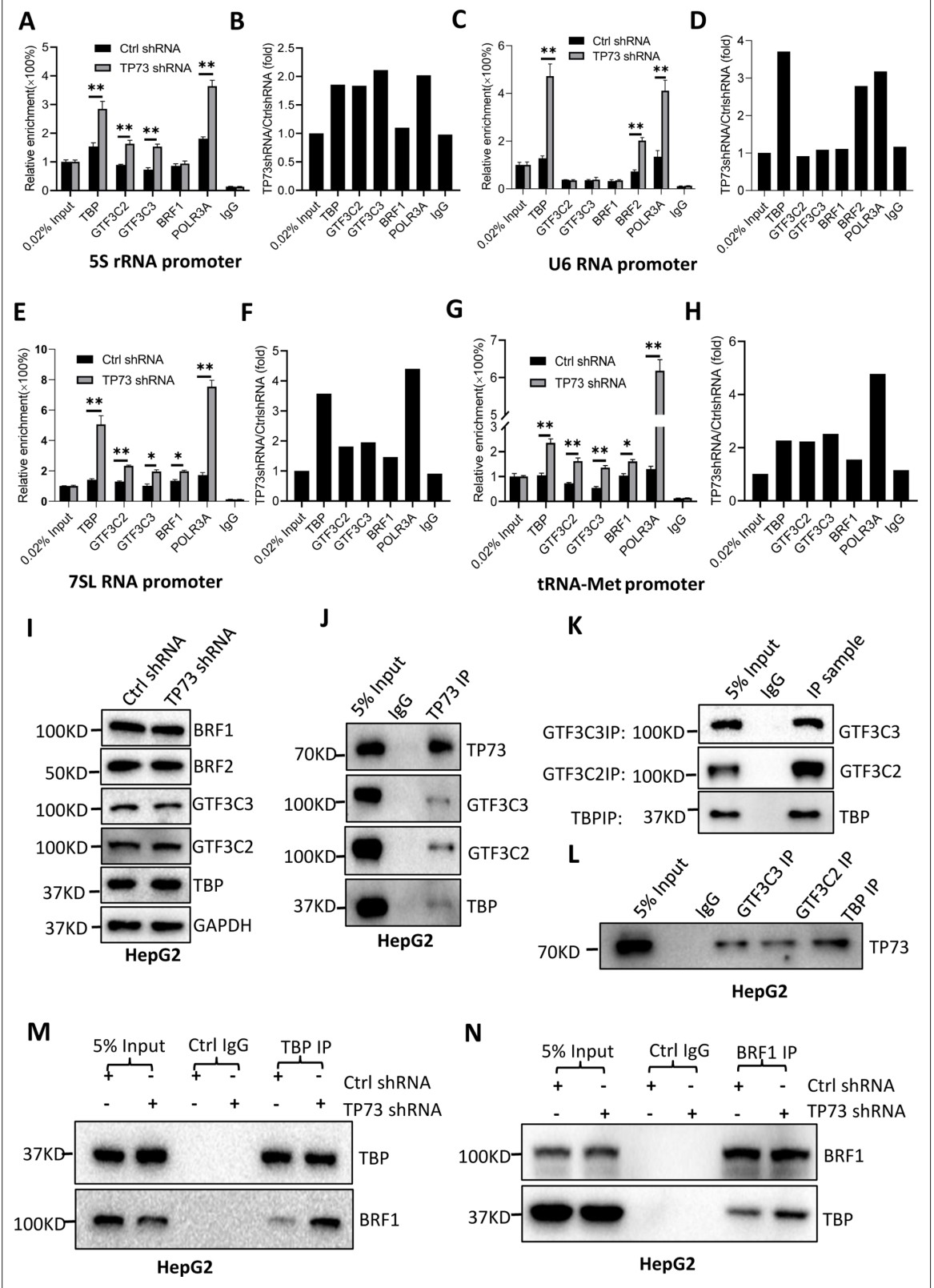

**Figure 6.** TP73 suppresses Pol III-directed transcription by disrupting TFIIIB assembly. (**A–H**) TP73 silencing increased the recruitment of the Pol III transcription machinery components to Pol III target loci. Chromatin immunoprecipitation (ChIP) assays were performed using HepG2 cell lines stably expressing TP73 shRNA or control shRNA. Relative enrichment (**A, C, E, and G**) was obtained by comparing the quantity of target DNA in 1 µL of ChIP DNA sample (1/40) to that in 0.02% of input DNA (1 ng genomic DNA). Fold change (**B, D, F, and H**) was obtained by comparing the mean of

*Figure 6 continued on next page*

*Figure 6 continued*

the relative enrichment of target DNA in TP73 shRNA samples to that in control shRNA samples. (**I**) Western blot images showing the effect of TP73 depletion on the expression of Pol III transcription machinery components. (**J**) Western blot images showing the results of TP73 co-immunoprecipitation (co-IP) assays. Co-IP assays were performed using an anti-TP73 antibody and HepG2 nuclei extract. (**K and L**) Western blot images showing the results of co-IP assays with antibodies against the indicated factors in K. The immunoprecipitation efficiency between antibody and antigen was verified by western blot (**L**). (**M**) TP73 silencing increased TBP binding to BRF1. Co-IP assays were performed using HepG2 cells expressing TP73 shRNA or control shRNA and an anti-TBP antibody. (**N**) TP73 silencing increased BRF1 binding to TBP. Co-IP assays were performed using HepG2 cells expressing TP73 shRNA or control shRNA and an anti-BRF1 antibody. Each column in A, C, E, and G represents the mean of SD of three biological replicates (n=3). *, p<0.05; **, p<0.01. p Values were obtained by Student's *t* test performed with the data from control and treated samples. Mean, SD, and p values were calculated using GraphPad Prism 8 software.

The online version of this article includes the following source data and figure supplement(s) for figure 6:

**Source data 1.** Raw images for Western blot data in *Figure 6*.

**Source data 2.** Original digital data for *Figure 6*.

**Figure supplement 1.** TP73 knockdown increased the interaction between TBP and BRF2.

**Figure supplement 1—source data 1.** Raw images for Western blot data in *Figure 6—figure supplement 1*.

survival probability and shorter survival time compared to those with low expression levels of miR-106a-5p (*Figure 7I*). Using the same method, the relationship between miR-106a-5p expression level and survival probability for kidney cancer patients was analyzed based on the TCGA database; the result is consistent with that obtained with liver cancer data (*Figure 7J*). These results suggest that miR-106a-5p can act as a biomarker for a subset of cancer prognosis, including liver and kidney cancers.

## miRNA-106a-5p inhibits TP73 mRNA and is required for Pol III-directed transcription

Bioinformatics analysis showed that miR-106a-5p potentially binds to the TP73 mRNA 3'UTR. To support this probability, we synthesized DNA fragments encoding the TP73 mRNA 3'UTR recognized by miR-106a-5p (TP73 3'UTR WT), and the resulting fragments were inserted downstream of a *luciferase* gene within the reporter vector pmirGLO. Meanwhile, a DNA fragment mutant disrupting miR-106a-5p binding to the TP73 mRNA 3'UTR (TP73 3'UTR MT) was inserted into the same vector in parallel (*Figure 8A*). The resulting vectors and miR-106a-5p mimics were co-transfected into 293T and HepG2 cells. Luciferase activity was assessed using transiently transfected cells and a luciferase detection kit. The co-transfection of miR-106a-5p mimics and reporter vectors containing the TP73 3'UTR WT reduced luciferase activity, whereas the co-transfection of miR-106a-5p mimics and reporter vectors containing the TP73 3'UTR MT did not significantly affect luciferase activity (*Figure 8B and C*). These results indicate that miR-106a-5p can recognize the TP73 3'UTR to interfere with *luciferase* gene expression. To understand whether miR-106a-5p regulates Pol III-directed transcription by affecting endogenous TP73 expression, we transfected 293T and HepG2 cells with miR-106a-5p mimics or control miRNA mimics. TP73 protein and Pol III products were monitored by western blot and RT-qPCR, respectively. Intriguingly, transfection of miR-106a-5p mimics severely reduced TP73 expression (*Figure 8D and F*) but significantly enhanced the expression of Pol III products (*Figure 8E and G*), indicating that miR-106a-5p can activate Pol III-directed transcription by inhibiting TP73 expression. We next determined whether miR-106a-5p is required for the regulation of Pol III-dependent transcription mediated by STAT3. To this end, we transfected miR-106a-5p mimics into the HepG2 cell line with STAT3 depletion. TP73 expression was detected by western blot using the resulting transfected cells. As expected, STAT3 shRNA stable expression enhanced TP73 expression; however, transfection of miR-106a-5p mimics inhibited the activation of TP73 expression induced by STAT3 silencing (*Figure 8H*). Analysis of Pol III products and cell proliferation revealed that transfection of miR-106a-5p mimics reversed the inhibition of Pol III product expression and proliferative activity induced by STAT3 silencing, indicating that miR-106a-5p is required for the regulation of Pol III-mediated transcription mediated by STAT3. Taken together, miR-106a-5p can inhibit TP73 expression by binding to the TP73 mRNA 3'UTR, through which the miR-106a-5p regulates Pol III-dependent transcription.

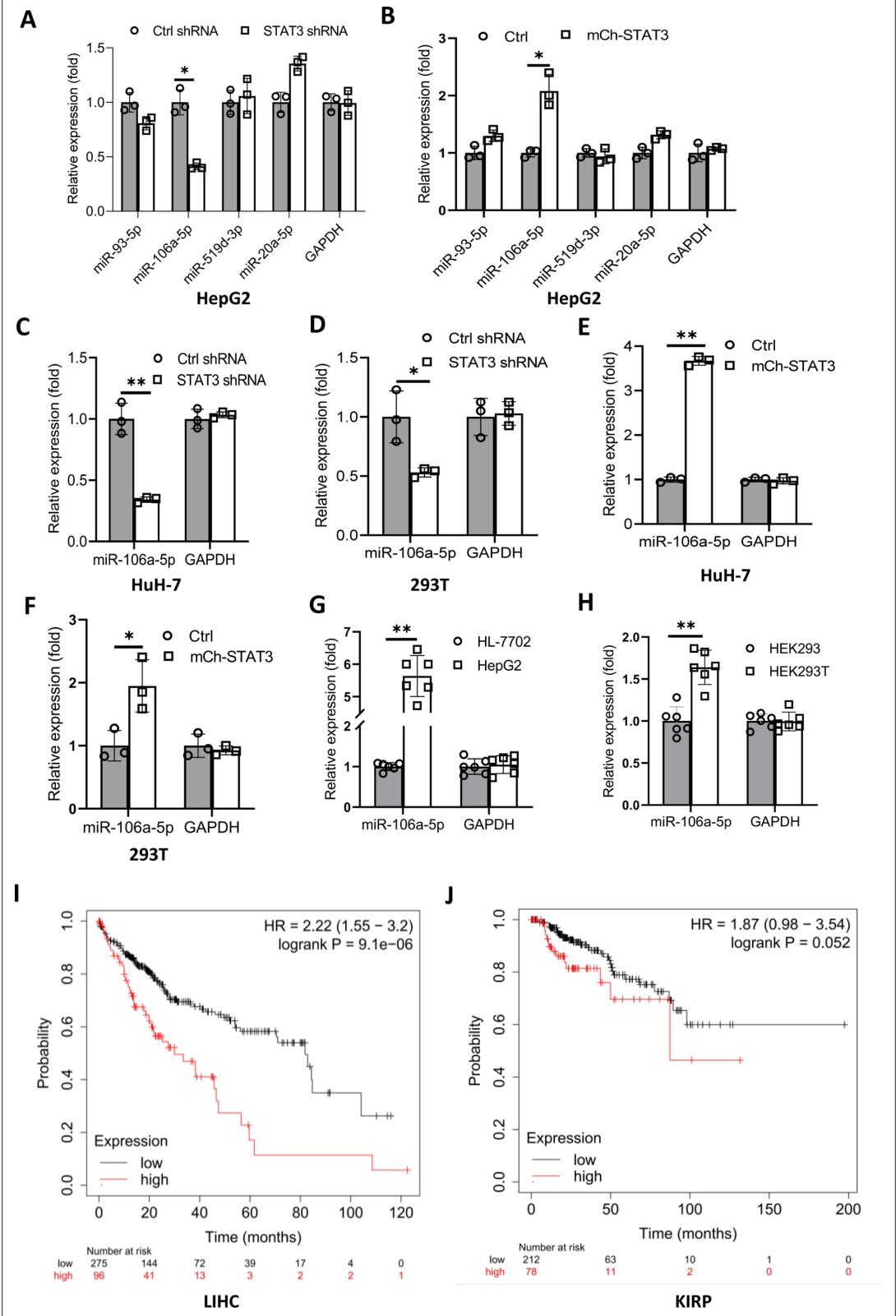

**Figure 7.** STAT3 positively regulates the expression of miR-106a-5p potentially targeting TP73 mRNA 3' UTR. (**A and B**) Alteration of STAT3 expression affected miR-106a-5p expression but did not affect the expression of other miRNA molecules, including miR-93–5 p, miR-519d-3p, and miR-20a-5p. RT-quantitative PCR (qPCR) was performed using HepG2 cell lines with STAT3 silencing or overexpression. (**C and D**) STAT3 depletion reduced the expression of miR-106a-5p in HuH-7 (**C**) and 293T (**D**) cells. RT-qPCR was performed using the cell lines expressing STAT3 shRNA or control shRNA. (**E**

*Figure 7 continued on next page*

*Figure 7 continued*

**and F)** STAT3 overexpression enhanced the expression of miR-106a-5p in HuH-7 (**E**) and 293T (**F**) cells. RT-qPCR analysis was performed using HuH-7 or 293T cell lines with STAT3 overexpression and the corresponding control cell lines. (**G and H**) Transformed cell lines showed the increased expression of miR-106a-5p compared to their corresponding normal cell lines. HepG2 and HEK293T cell lines and their corresponding normal cell lines, including HL-7702 (**G**) and HEK293 (**H**) were cultured and harvested for the analysis of miR-106a-5p expression by RT-qPCR. (**I and J**) Cancer patients with high expression of miR-106a-5p exhibit low-survival probability and shorter-survival time compared to those with low expression of miR-106a-5p in both liver hepatocellular carcinoma (LIHC, **I**) and kidney renal papillary cell carcinoma (KIRP (Kidney renal papillary cell carcinoma), **J**). Each column in A–H represents the mean of SD of three biological replicates (n=3). *, p<0.05; **, p<0.01. p Values were obtained by Student's *t* test performed with the data from control and treated samples. Mean, SD, and p values were calculated using GraphPad Prism 8 software.

The online version of this article includes the following source data and figure supplement(s) for figure 7:

**Source data 1.** The predicted top score miRNAs targeting TP73 mRNA.

**Source data 2.** Original digital data for *Figure 7*.

**Figure supplement 1.** Schemes showing discovery and detection of miR-106a-5p.

## STAT3 regulates miR-106a-5p gene transcription by binding to the miRNA promoter

To understand how STAT3 regulates miR-106a-5p expression, we searched and analyzed the transcription factor binding motif within the DNA sequence upstream of the *miR-106a-5p* gene. As expected, the *miR-106a-5p* gene promoter contains several STAT3 binding motifs (***Figure 9—figure supplement 1***). To verify whether STAT3 binds to the promoter of *the miR-106a-5p* gene, we performed ChIP assays using HepG2 cells and an anti-STAT3 antibody. ChIP qPCR results showed that STAT3 showed specific binding to the promoter of *the miR-106a-5p* gene (***Figure 9A***). Since STAT3 can bind to the *miR-106a-5p* promoter, we asked whether alteration of STAT3 expression affects transcription of *the miR-106a-5p* gene. To answer this question, we amplified the DNA fragment comprising the miR-106a-5p gene promoter and *miR-106a-5p* gene by PCR, and the resulting DNA fragments were inserted into pGL3-basic (***Figure 9B***). Next, transient transfection was performed using the promoter-driven reporter vectors and HepG2 cells with STAT3 depletion or overexpression; luciferase activity was analyzed using the resulting cells and a luciferase detection kit. Analysis of luciferase activity showed that STAT3 silencing reduced the luciferase activity (***Figure 9C***). In contrast, STAT3 overexpression enhanced the luciferase activity (***Figure 9D***). Since the *miR-106a-5p* gene was also incorporate into the pGL3-basic with the promoter. Thus, exogenous miR-106a-5p expression was assessed by RT-qPCR and the primers as indicated (***Figure 9B***). The results are in agreement with those obtained in luciferase assays (***Figure 9E and F***). These results indicate that STAT3 can regulate the *miR-106a-5p* gene transcription by binding to the *miR-106a-5p* promoter.

Based on the results obtained in this study, we proposed a model by which STAT3 positively regulates Pol III-directed transcription. Specifically, STAT3 first activates miR-106a-5p expression by directly binding to its promoter, and the activation of miR-106a-5p leads to the suppression of TP73 expression. Since TP73 interacts with TFIIIB subunit TBP, TP73 downregulation can enhance TFIIIB assembly at Pol III-transcribed loci by reducing interaction with TBP, which subsequently activates Pol III-dependent transcription (***Figure 9G***).

## Discussion

In this study, we found that STAT3 can positively regulate Pol III-directed transcription (***Figure 1***, ***Figure 1—figure supplement 1***). STAT3 was shown to regulate RNA polymerase II-mediated transcription, including *CyclinD1*, *Survivin*, *VEGF*, *MMP2/9*, and *NF-kB* gene transcription (***Srivastava and DiGiovanni, 2016***; ***Lee et al., 2019***). Thus, the findings from this study extend the roles of STAT3 in transcriptional regulation. TP73, a member of the p53 family, is expressed as two isoforms: transcriptionally active (TA) and inactive forms (ΔN). It has been shown that ΔN TP73 has oncogenic potential and exhibits a dominant negative behavior to TP73 and p53 (***Maas et al., 2013***; ***Ozaki and Nakagawara, 2005***). TP73 has also been found to regulate skin development and plays a role in skin carcinogenesis (***Botchkarev and Flores, 2014***). Interestingly, we showed that STAT3 activates TP73 expression at both RNA and protein levels. TP73 not only independently inhibits Pol III-directed transcription but also participates in the inhibition of Pol III-dependent transcription induced by STAT3

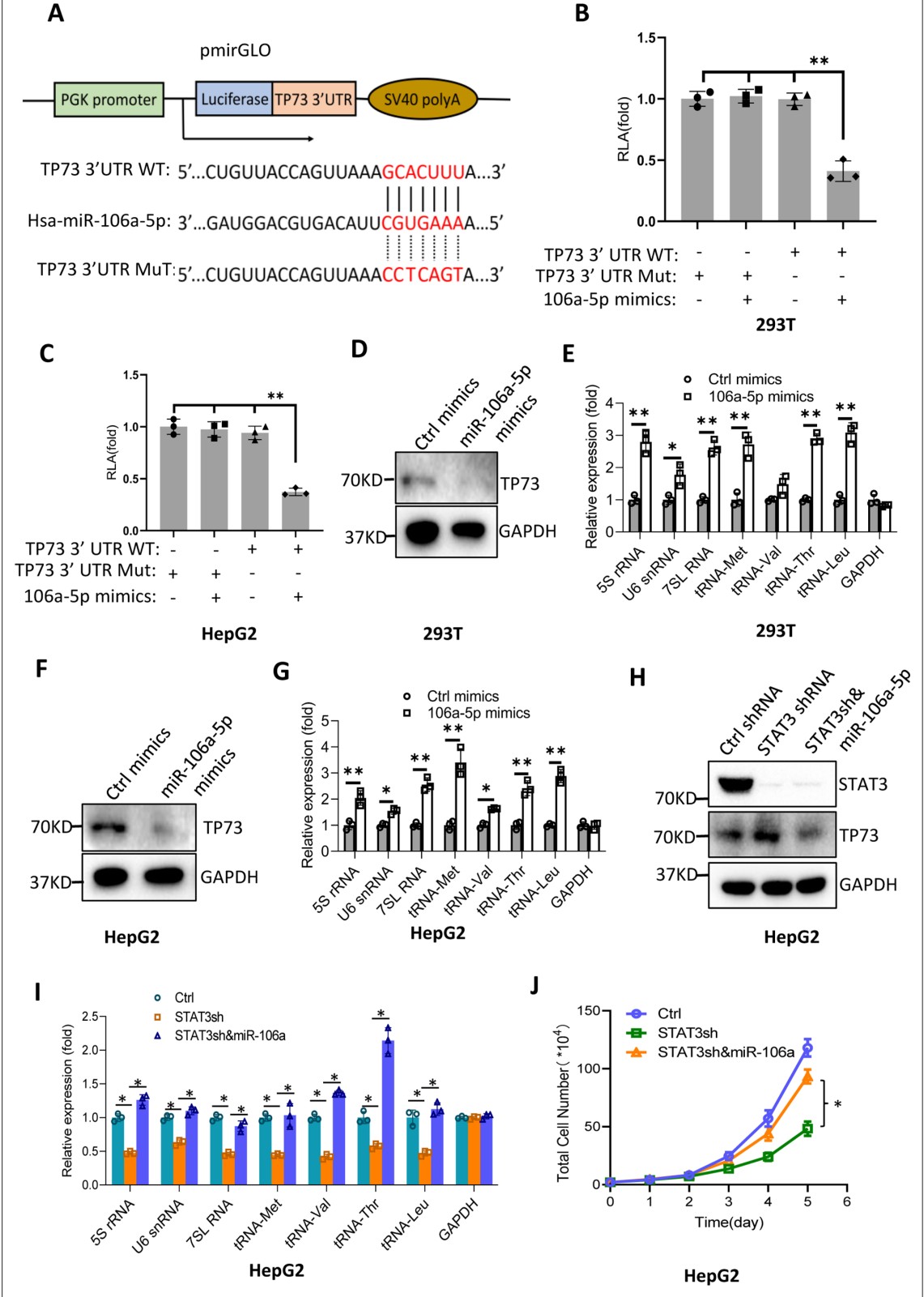

**Figure 8.** MiR-106a-5p activates Pol III-directed transcription by inhibiting TP73 expression. (**A**) A diagram showing cloning of the TP73 3'UTR and the complementary region between miR-106a-5p and the wild-type TP73 3'UTR (WT) or the TP73 3'UTR mutant (MuT). (**B and C**) Co-transfection of miR-106a-5p mimics and reporter expression vectors containing a DNA fragment encoding the WT TP73 3'UTR decreased luciferase activity in 293T (**B**) and HepG2 cells (**C**). (**D–G**) Transfection of miR-106a-5p mimics activated the expression of Pol III products in both 293T (**D and E**) and HepG2 cells (**F and**

*Figure 8 continued on next page*

*Figure 8 continued*

**G**). TP73 and Pol III products were detected by western blot (**D and F**) and RT-qPCR (**E and G**), respectively. (**H**) Western blot results for HepG2 cells expressing STAT3 shRNA with or without transfection of miR-106a-5p mimics. (**I**) Transfection of miR-106a-5p mimics reversed the inhibition of Pol III-dependent transcription induced by STAT3 silencing. (**J**) Transfection of miR-106a-5p mimics reversed the inhibition of HepG2 cell proliferation induced by STAT3 silencing. Each column or point in histograms represents the mean of SD of three biological replicates (n=3). *, p<0.05; **, p<0.01. p Values were obtained by Student's *t* test (**B, C, E, G, and I**) or two-way ANOVA (**J**). Mean, SD, and p values were calculated using GraphPad Prism 8 software.

The online version of this article includes the following source data for figure 8:

**Source data 1.** Raw images for Western blot data in *Figure 8*.

**Source data 2.** Original digital data for *Figure 8*.

silencing (*Figures 4–6*), suggesting that STAT3 modulates Pol III-directed transcription by controlling TP73 expression. Therefore, in this study, we have also identified a novel role of TP73 in Pol III-directed transcription. Coincidently, p53 has been reported to repress Pol III-directed transcription by interacting with TBP (*Crighton et al., 2003*; *Cairns and White, 1998*), and p53 does not disrupt the interaction between TBP and BRF1 rather than prevents TFIIIB association with TFIIIC2 and RNA pol III (*Crighton et al., 2003*). However, we showed that TP73 can interact with TBP, and TP73 silencing increased the interaction between TBP and BRF1 (*Figure 6J–N*), suggesting that TP73 inhibits Pol III-directed transcription by disrupting the assembly of TFIIIB subunits. Thus, the regulatory mechanism of Pol III transcription mediated by TP73 is distinct from that mediated by p53, but it resembles the regulatory mechanism of Pol III-directed transcription mediated by PTEN/AKT, where PTEN disrupts the association of TBP and BRF1 (*Woiwode et al., 2008*).

miRNA-106a-5p has been shown to regulate a variety of biological processes, including cell proliferation, migration, autophagy, and ferroptosis (*Zhou et al., 2021*; *Zhang et al., 2021*). MiR-106a-5p can function as a tumor suppressor by targeting a vascular endothelial growth factor (VEGF) in renal cell carcinoma (*Ma et al., 2020*). In a recent study, miR-106a-5p was found to be upregulated in spine cord glioma compared to normal tissues, and the inhibition of miR-106a-5p blunts cell proliferation, migration, and invasion (*Xu et al., 2021*), suggesting that miR-106a-5p positively regulates cancer development. In the present study, we found that transfection of miR-106–5p mimics can inhibit TP73 protein expression by targeting the TP73 mRNA 3'UTR and activating Pol III-directed transcription. STAT3 positively regulates miR-106–5p expression by binding to the *miR-106a-5p* promoter (*Figures 8 and 9*). Thus, miR-106a-5p links STAT3 with TP73 and Pol III-directed transcription; this is another novel finding achieved in the present work.

We showed that either STAT3 or TP73 expression alteration can affect the synthesis of Pol III products and cell proliferation. The presence of ML-60218 (a Pol III transcription inhibitor) inhibits Pol III-dependent transcription and cell proliferation (*Figure 2*, *Figure 2—figure supplements 1 and 2*, *Figure 5—figure supplement 3*), suggesting that STAT3 promotes cell proliferation by activating Pol III-directed transcription, although the contribution of other factors or pathways to cell proliferation enhancement cannot be excluded. Indeed, in our latest study, we found that STAT3 also drives Pol I-directed transcription by activating RPA34 expression, and both STAT3 and Pol I transcription-specific inhibitors can suppress cancer cell proliferation (*manuscript accepted*). In addition, STAT3 has been shown to promote cell proliferative activity in several types of cancer stem cells (*Canesin et al., 2020*; *Park et al., 2019*). The inhibition of Pol III activity by depleting POLR3G in prostate cancer cells suppresses cancer stem cell proliferation and initiates cell differentiation (*Petrie et al., 2019*). Therefore, our findings together with the previous findings provide an excellent opportunity to develop novel anti-cancer drugs by inhibiting the activities of STAT3, Pol I, and Pol III products. Moreover, inhibitors against STAT3, Pol I, and Pol III transcription would be combined and used in anti-cancer research in the future.

Taken together, in this study, we found that STAT3 positively modulate Pol III-directed transcription and cell proliferation. STAT3 inhibits TP73 expression and activates miR-106–5a expression. Both miR-106a-5p and TP73 are not only required for the regulation of Pol III-directed transcription mediated by STAT3 but also independently regulate Pol III-dependent transcription. Transfection of miR-106a-5p mimics was able to reduce TP73 expression by targeting TP73 mRNA 3'UTR and activate Pol III-dependent transcription. Therefore, STAT3 modulates Pol III-directed transcription by controlling the miR-106a-5p/TP73 axis. The findings from this study shed light on the regulatory mechanism of Pol III-directed transcription and lay a foundation for developing novel drugs for cancer therapy.

**Figure 9.** STAT3 positively regulates miR-106a-5p transcription by binding to the *miR-106a-5p* promoter. (**A**) Chromatin immunoprecipitation (ChIP)-quantitative PCR (qPCR) results showing STAT3 binding to the *miR-106a-5p* promoter. ChIP assays were performed using HepG2 cells and an anti-STAT3 antibody. Relative enrichment was obtained as described in *Figure 3A*. (**B**) A scheme showing the cloning of the DNA fragment comprising the *miR-106a-5p* promoter and a *miR-106a-5p* gene. (**C and D**) Effect of STAT3 upregulation and downregulation on the activity of the *miR-106a-5p* promoter. HepG2 cell lines with STAT3 silencing (**C**) and overexpression (**D**) were transfected using the promoter-driven reporter vectors. Luciferase assays were performed using the cell lysate from transfected cell lines. (**E and F**) Effect of STAT3 upregulation and downregulation on the expression of exogenous miR-106a-5p. HepG2 cell lines with STAT3 silencing (**E**) and overexpression (**F**) were transfected with the promoter-driven reporter vectors. The expression of exogenous miR-106a-5p was detected by RT-qPCR using the primers targeting miR-106a-5p (FP) and luciferase mRNA (RP) as indicated in B. (**G**) A proposed model by which STAT3 regulates Pol III-directed transcription. Each column in A and C–F represents the mean of SD of three biological replicates (n=3). **, p<0.01. p Values were obtained by Student's *t* test. Mean, SD, and p values were calculated using GraphPad Prism 8 software.

The online version of this article includes the following source data and figure supplement(s) for figure 9:

*Figure 9 continued on next page*

*Figure 9 continued*

**Source data 1.** Original digital data for *Figure 9*.

**Figure supplement 1.** A DNA sequence that contains putative STAT3 binding sites (green bold bases) and a *miR-106a-5p* gene (red bold bases).

## Materials and methods

### Gene cloning, cell culture, and reagents

The shRNA-expressing lentiviral plasmid pLV-U6-EGFP-Puro was obtained from Inovogen Tech Co (Beijing, China). Three dsDNA fragments encoding distinct STAT3 shRNAs were inserted downstream of the U6 promoter within the pLV-U6-EGFP-Puro plasmid. STAT3 cDNA was synthesized using the total RNA extracted from 293T cells and reverse transcriptase (New England Biolabs, USA), amplified by PCR, and inserted immediately downstream of a *mCherry* gene in the protein-expressing lentiviral plasmid pLV-EF1α-mCherry-puro. HepG2 (ATCC HB8065, USA), HuH-7 (NCACC, SCSP-526, Shanghai, China), and 293T (ATCC CRL-3216, USA) cell lines were cultured in their respective media supplied with 10% fetal bovine serum (Gibcol Co.) and 1×penicillin/streptomycin (Hyclone Co). These cell lines have been authenticated by STR profiling test and proved without contamination before used in various assays in the present study. Biological reagents, including transfection reagents, qPCR Master Mix, ECL western blot detection solution, and others, were purchased from Thermo Scientific. Chemical reagents were obtained from Sinopharm Chemical Reagent Co (Shanghai, China).

### Transfection and generation of cell lines

For the generation of cell lines stably expressing shRNA or fusion protein, lentiviral vectors expressing STAT3 or TP73 shRNA or expressing mCherry-STAT3 were transiently transfected into 293T cells along with the packaging plasmids pH1 and pH2. After 48 hr, the culture medium containing lentiviral particles was collected and filtered with a sterilized filter (0.45 μm). Meanwhile, HepG2, HuH-7, and 293T cells were grown in 12-well plates; at 70% confluence, cells were incubated with the lentiviral particle-containing medium. The cell lines stably expressing shRNA or exogenous protein were obtained by selecting with puromycin (5 μg/mL) and screening using 96-well plates. The colonies with shRNA or fusion protein stable expression were verified by western blot using antibodies against STAT3, TP73, and mCherry.

### Real-time qPCR

HepG2 or other cell lines stably shRNA or fusion protein were cultured in 3.5 cm dishes. At 85% confluence, cells were harvested, and total RNA was extracted from the cells using an Axygen RNA extraction kit. After quantification, 0.25 μg of the total RNA was used for cDNA synthesis within 20 μL reaction mixtures according to the protocol provided by a reverse transcription kit (Thermo Fisher Scientific). When the reaction finished, the resulting products were diluted to 100 μL with ddH$_2$O. For one qPCR reaction, 1 μL of cDNA samples was mixed with 0.5 μL of 5 μM forward and reverse primers and 10 μL of 2×SYBRGreen PCR Mastermix within a 20 μL reaction system. qPCR reactions were performed in a Bio-Rad real-time detection system as described previously (*Zhang et al., 2022*). An *Actin* gene was used as a reference gene, and *GAPDH* gene expression acted as a negative control when qPCR data were processed. The sequences of all primers used in qPCR were presented in *Supplementary file 1*.

### The analysis of cell proliferation and colony formation

Cell proliferative activity was detected using several distinct methods, including cell counting, cell counting kit-8 (CC-8), and EdU assays. For cell counting assays, HepG2, HuH-7, and 293T cell lines with STAT3 silencing or overexpression and their corresponding control cell lines were cultured in 12-well plates. Cell counting was performed every 24 hr, and the resulting data were subjected to statistical analysis. For CCK-8 assays, cells were seeded in 96-well plates, and cell proliferation was monitored based on the protocol as described previously (*Wu et al., 2022*). For EdU assays, cell proliferation was assessed using the protocol as described previously (*Wu et al., 2022*). The rate of EdU-positive cells was obtained by comparing the number of the EdU-positive cells to that of the total cells. For proliferation assays with or without ML-60218, a HepG2 cell line with STAT3 overexpression and its control cell line were cultured in 12-well plates, where samples from each cell line were divided

into two groups; one group of samples were cultured in the medium containing 54 µM ML20618. Cell samples in each group were harvested, and cell counting was performed cell proliferation every 24 hr, whereas Pol III product expression was analyzed by RT-qPCR using cell samples cultured for 48 hr.

Colony formation assays were performed using HepG2 cells cultured in 6-well plates. Briefly, around $1\times10^3$ HepG2 cells with STAT3 silencing or overexpression and their corresponding control cells were seeded in 6-well plates, and the culture medium was replaced every 2 days. After growing for 12 days, cells were fixed with ice-cold 100% methanol and stained with 0.3% crystal violet. After washing three times with deionized distilled $H_2O$, cell colonies were pictured with a camera.

## Mouse models for tumor formation

12 of the 5-week-old female BALB/C nude mice were purchased from the Vital River Laboratory Animal Technology Co (Beijing, China). These nude mice were nurtured under a sterile condition with controlled temperature, humidity, and light. After inhabiting for 1 week, nude mice (n=6 for each group) were injected subcutaneously with $1\times10^7$ HepG2 cells expressing STAT3 shRNA or control shRNA. After growing 7 days, tumors within nude mice appeared and were measured with a vernier caliper every 7 days. Tumor volumes were calculated using the following formula: $V = \frac{\pi}{6} \times$ length $\times$ width$^2$. At the end of the fifth week, nude mice were euthanized, and tumors within mice were removed, weighed, and photographed. The resulting data were subjected to statistical analysis. Tumor samples randomly picked from controls or treatments were used for immunohistochemistry analysis performed by the Analysis Center at Wuhan University of Science and Technology. Animal experiments were approved by the Animal and Medical Ethics Committee of the School of Life Science and Health at the Wuhan University of Science and Technology. The animal protocols abided by the Animal Welfare Guidelines (China), and the approval with a stamp and reference number (WKDLL-2020–019) is available on request.

## ChIP assays

HepG2 cells with STAT3 or TP73 silencing and their control cells were cultured in 10 cm dishes. At 85% confluence, cells were fixed for 10 min using 10 mL of 1% formaldehyde solution freshly prepared with 1×PBS and subsequently quenched with 1 mL of 2.5 M glycine solution. After washing three times with 1×PBS solution, cell samples were harvested with cell scrappers and were sheared with an ultrasonication machine (Scientz 98-III,Ningbo, China). After sonicating for 20 min (pulse 5 s and interval 5 s) with 25% power, cell lysate was centrifuged for 10 min at 12,000 rpm, and the resulting supernatant was retained for ChIP assays as described previously (*Zhang et al., 2022*), where antibodies against TBP (SC-204; Santa Cruz Biotech.), BRF1 (SC-81405; Santa Cruz Biotech.), GTF3C2 (SC-81406; Santa Cruz Biotech.), and POL3RA (SC-292119, Santa Cruz Biotech.) were used in ChIP. The DNA from ChIP assays was purified with a QIAGEN PCR clean kit and eluted with 40 µL ddH$_2$O. 1 µL of ChIP DNA sample (1/40) was used for a qPCR reaction, while 0.02% input DNA (0.1 ng genomic DNA) was used in a qPCR reaction for positive control. Relative enrichment was obtained by comparing the quantity of target DNA in 1 µL of ChIP DNA sample to that in 0.02% input DNA.

## Western blot analysis

HepG2, HuH-7, and 293T cells were cultured in 6-well plates. At 90% confluence, cells were harvested and lyzed with 1×SDS loading buffer. After heating at 100°C for 10 min, 20 µL cell lysate was loaded for SDS-PAGE electrophoresis and western blot analysis. Western blot was performed using antibodies against STAT3 (CST#9139, CST), mCherry (TAG0080, Frdbio), GAPDH (RAB0101, Frdbrio), BRF1 (SC-81405, Santa Cruz Biotech), GTF3C2 (SC-81406, Santa Cruz Biotech), GTF3C3 (SC-393235, Santa Cruz Biotech), TBP (SC-421, Santa Cruz Biotech), and TP73 (CSB-PA003696, CUSABio).

## Co-IP analysis

HepG2 cells were cultured in 10 cm dishes. At 90% confluence, cell samples were harvested, and nuclei were isolated using a hypotonic buffer (20 mM Tris-HCl pH 7.4, 10 mM NaCl, and 3 mM MgCl2) and a NP-40 buffer (20 mM Tris-HCl pH 7.4, 10 mM NaCl, 3 mM MgCl2, and 0.5% NP-40). After isolation, nuclei were suspended using a modified nuclear extraction buffer (25 mM Tris-HCl pH 7.4, 300 mM NaCl, 1 mM DTT, 1% triton-100, 0.1% SDS, and 2 mM PMSF) and subjected to vortexes for 30 min. The nuclear suspension was centrifuged for 10 min at 13,000 rpm, and the supernatant

(nuclear extract) was retained for IP assays. IP reactions were performed using 200 µL nuclear extract and 5 µg of antibodies, respectively, against TP73 (CSB-PA003696, CUSABio), TBP (SC-421, Santa Cruz Biotech), BRF1 (SC-81405, Santa Cruz Biotech), GTF3C2 (SC-81406, Santa Cruz Biotech), and GTF3C3 (SC-393235, Santa Cruz Biotech). Antibody-bound proteins were precipitated by incubating with protein A/G agarose and washed with a modified RIPA buffer (0.05%SDS, 0.1% sodium deoxycholate, 1% Triton X-100, 1 mM EDTA, 0.5 mM EGTA, 300 mM NaCl, and 10 mM Tris-HCl, pH 8.0). IP samples were eventually eluted with 40 µL of 1×SDS loading buffer (50 mM Tris-HCl, 2% SDS, 0.1% bromophenol blue, 10% glycerol, and 100 mM DTT). Western blot was performed using 10 µL of the IP samples, where 5% input was loaded as a positive control. The antibodies used for western blot were the same as those used for IP assays.

## Messenger RNA-seq analysis

A HepG2 cell line expressing STAT3 shRNA and its control cell line established above were cultured in 10 cm dishes in triplicates. At 85% confluence, cells were harvested, and total RNA was prepared using a Axygen RNA minprep kit. After quantification, the total RNA was sent for mRNA-seq analysis. The clean data were aligned to the human reference genome (Hg38) using Hisat2 software (version 2.0.1, Daehwankim lab). DEGs were analyzed using the DESeq Bioconductor package as described previously (*Anders and Huber, 2010*). p Values for the DEGs were set at less than 0.05. GO-Term-Finder was used to identify gene ontology terms that annotate a list of enriched genes where their p values were less than 0.05. The heatmap of the DEGs between control shRNA and STAT3 shRNA samples was obtained in R with the pheatmap package. The volcano plot for up- or downregulation genes was obtained with the ggplat2 package (V3.3.6).

## Reporter assays

DNA fragments that encode the wild-type TP73 mRNA 3'UTR recognized by miR-106A-5p were synthesized and inserted downstream of a luciferase gene in the reporter vector pmiRGLO. Meanwhile, a mutant disrupting miR-106A-5p binding was generated in parallel. The resulting vectors and miR-106a-5p mimics were co-transfected into 293T and HepG2 cells cultured in 12-well plates. After 48 hr, cells were harvested and lysed with 100 µL of lysis buffer provided by a luciferase detection kit (Promega). After centrifugation, the supernatant was retained, and 5 µL of the supernatant was used to analyze the activities of luciferase. Relative luciferase activity was obtained by comparing the luciferase activity of the cells co-transfected with both miRNA mimic and the reporter vectors containing the wild-type TP73 3'UTR to that from the cells transfected with the reporter vector containing the mutated TP73 3' UTR.

## Kaplan-Meier plotting and statistical analysis

Kaplan-Meier plotting showing the relationship between miR-106a-5p and survival probability and time was performed using the Kaplan-Meier Plotter online tool (https://www.kmplot.com), which is based on the RNA-seq data of LIHC and kidney renal carcinoma deposited at the TCGA. The experiments in this study were performed with minimal three biological replicates (n≥3). The mean and SD for the data from gene expression analysis, cell proliferation assays, ChIP assays, and luciferase assays were calculated with the GraphPad Prism 8.0 software. Each bar/point in the histograms represents the mean ± SD (n≥3). p Values were obtained by Student's *t* test or two-way ANOVA.

## Acknowledgements

This work was funded by a project of the Natural Science Foundation of China (31671357 to WD, 82102888 to HD).

## Additional information

### Funding

| Funder | Grant reference number | Author |
| --- | --- | --- |
| National Natural Science Foundation of China | 31671357 | Wensheng Deng |
| National Natural Science Foundation of China | 82102888 | Huan Deng |

The funders had no role in study design, data collection and interpretation, or the decision to submit the work for publication.

### Author contributions

Cheng Zhang, Formal analysis, Validation, Investigation, Methodology; Shasha Zhao, Data curation, Supervision, Methodology, Project administration; Huan Deng, Data curation, Supervision, Project administration; Shihua Zhang, Software, Formal analysis, Methodology; Juan Wang, Resources, Validation, Methodology; Xiaoye Song, Data curation, Validation, Methodology; Deen Yu, Resources, Software, Formal analysis; Yue Zhang, Resources, Validation; Wensheng Deng, Conceptualization, Funding acquisition, Writing - original draft, Writing - review and editing

### Author ORCIDs

Wensheng Deng (iD) http://orcid.org/0000-0003-2454-2288

### Ethics

Animal experiments were approved by the Animal and Medical Ethics Committee of the School of Life Science and Health at the Wuhan University of Science and Technology. The animal protocols abided by the Animal Welfare Guidelines (China), and the approval with a stamp and reference number (WKDLL-2020-019) is available on request.

### Decision letter and Author response

Decision letter https://doi.org/10.7554/eLife.82826.sa1
Author response https://doi.org/10.7554/eLife.82826.sa2

## Additional files

### Supplementary files

- Supplementary file 1. A table containing sequences for RT-qPCR primers and miRNAs.
- MDAR checklist

### Data availability

1. The RNA-seq data for HepG2 cell lines expressing STAT3 shRNA or Control shRNA were deposited in the NGDC Genome Sequence Archive for human (HRA002939, https://ngdc.cncb.ac.cn/gsa-human/browser/HRA002939). 2. The Kaplan Meier plotting between miRNA expression and cancer patient survival time were obtained by the following website: http://kmplot.com/analysis/index.php?p=service&cancer=pancancer_mirna. 3. WB source data were used to generate Western blot images in main figures and figure supplements. 4. Histogram source data contain numerical data were use to generate all histograms in main figures and figure supplements.

The following dataset was generated:

| Author(s) | Year | Dataset title | Dataset URL | Database and Identifier |
| --- | --- | --- | --- | --- |
| Deng W | 2022 | Transcriptome analysis of control and STAT3 knockdown in HepG2 cell lines | http://ngdc.cncb.ac.cn/gsa-human/browse/HRA002939 | NGDC-GSA for human, HRA002939 |

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
