## [Editor Report]

The author arrived at the convincing conclusion that STAT3 expression promotes TFIIIB assembly through miR-106A-5p-mediated inhibition of TP73 expression, thereby increasing Pol III transcription, which contributes to enhanced cell proliferation. The data are very good and clearly support the proposed model.

---

## [Decision Letter]

**Decision letter after peer review:**

Thank you for submitting your article "STAT3 promotes RNA polymerase III-directed transcription by controlling the miR-106a-5p/TP73 axis" for consideration by *eLife*. Your article has been reviewed by 2 peer reviewers, and the evaluation has been overseen by Michael Green as a Reviewing Editor and James Manley as the Senior Editor. The following individuals involved in the review of your submission have agreed to reveal their identity: Martin Teichmann (Reviewer #1); Robert J. White (Reviewer #2).

The reviewers have discussed their reviews with one another, and the Reviewing Editor has drafted this to help you prepare a revised submission. Overall the reviewers were quite positive about the suitability of this submission for publication in *eLife*. The reviewers have specified two revisions as essential for publication in *eLife*, along with several other recommended revisions, as detailed in the individual reviews.

Essential revisions:

1. The authors need to complete the partial characterization of U6 transcription as a function of STAT3 expression. These studies would only concern the ChIP and Western blot experiments of figures 3C and J, as well as 6C and I. The authors should include BRF2 in these experiments.

With these experiments, the authors could show that not only U6 transcription is affected by STAT3 and TP73 expression (Figures 1 and 5) but also that the association of BRF2 with the U6 gene is (or is not) affected. This would result in the presentation of comparable results to those they showed to characterize Pol III genes with gene-internal promoters. Without these experiments, the analysis of Pol III transcribed genes with promoters upstream of the transcription initiation site remains incomplete.

2. Petrie et al. (2019) demonstrated that pol III activity promotes the ability of cancer stem cells (CSC) to initiate tumours (Nucleic Acids Res 47, 3937). Given the well-established role of STAT3 in CSC generation, it would be worth discussing if pol III control contributes to the stimulatory effects of STAT3 on CSC.

*Reviewer #1 (Recommendations for the authors):*

To fully investigate the effects of STAT3 on Pol III transcription, the authors should determine whether TP73 also affects Pol III transcription of genes controlled by promoters located 5' of the transcription initiation site. To this end, they should monitor the expression of these genes and analyze the assembly of the BRF2-containing form of TFIIIB in response to STAT3 expression. U6 data were used as a negative control for Pol-III genes not regulated by BRF1-containing TFIIIB, but direct data on STAT3-dependent regulation of the U6 gene or other SNAPc-dependent Pol-III genes are lacking.

In addition, co-IP of recombinant TP73 and recombinant TFIIIB subunits (TBP, BDP1, BRF1, BRF2) should be performed to gain better mechanistic insight into which TFIIIB subunit TP73 interacts with in regulating Pol III transcription (presumably BRF1 and BRF2).

If the authors are able to address these questions, I am in favor of publication in *eLife*.

*Reviewer #2 (Recommendations for the authors):*

The Abstract states that "…the pathways and factors that control Pol III-directed transcription have yet to be elucidated." This fails to acknowledge the considerable amount of published work investigating many aspects of pol III transcriptional regulation. The current manuscript makes a very valuable addition, but it is building on a well-established field.

P8. The authors should explain what is measured by the CCK8 approach.

P11. Figure 3I presents ChIP data in which binding of STAT3 is not detected at several pol III-transcribed loci. The conclusion that STAT3 does not act directly at these sites would be greatly strengthened by a positive control confirming that STAT3 binding can clearly be detected at other loci in this assay.

P16. Figure 6I seems to show some change in levels of BRF1, GTF3C2, and GTF3C3 when TP73 is depleted, contrary to the authors' statement (lines 340-341).

Petrie et al. (2019) demonstrated that pol III activity promotes the ability of cancer stem cells (CSC) to initiate tumours (Nucleic Acids Res 47, 3937). Given the well-established role of STAT3 in CSC generation, it would be worth discussing if pol III control contributes to the stimulatory effects of STAT3 on CSC.

Line 362:

….STAT3 was originally identified as a positive regulator in RNA polymerase II-directed…

should be rephrased into:

….STAT3 was originally identified to be a positive regulator in RNA polymerase II-directed…

Line 405:

Change:

….To suppoort this probability…..

into

….To support this probability…..

---

## [Author Response]

Essential revisions:1. The authors need to complete the partial characterization of U6 transcription as a function of STAT3 expression. These studies would only concern the ChIP and Western blot experiments of figures 3C and J, as well as 6C and I. The authors should include BRF2 in these experiments.With these experiments, the authors could show that not only U6 transcription is affected by STAT3 and TP73 expression (Figures 1 and 5) but also that the association of BRF2 with the U6 gene is (or is not) affected. This would result in the presentation of comparable results to those they showed to characterize Pol III genes with gene-internal promoters. Without these experiments, the analysis of Pol III transcribed genes with promoters upstream of the transcription initiation site remains incomplete.

We totally agree with this comment. We have analyzed the effect of STAT3 expression alteration on BRF2 expression by Western blot and examined BRF2 binding at the U6 snRNA promoter by performing ChIP assays with BRF2 antibody, and we added the data into Figure 3C, J and K. Likewise, we analyzed the effect of TP73 expression alteration on BRF2 expression and performed CHIP assays using the BRF2 antibody, and the data obtained were added into Fig6 C and I. We very much thank the reviewers for the constructive comment.

2. Petrie et al. (2019) demonstrated that pol III activity promotes the ability of cancer stem cells (CSC) to initiate tumours (Nucleic Acids Res 47, 3937). Given the well-established role of STAT3 in CSC generation, it would be worth discussing if pol III control contributes to the stimulatory effects of STAT3 on CSC.

Yes, it is a good suggestion. We have discussed our findings by referring previous publications about the roles of STAT3 and Pol III products in cell stem cells growth and survival in the Discussion section of the revised manuscript.

Reviewer #1 (Recommendations for the authors):To fully investigate the effects of STAT3 on Pol III transcription, the authors should determine whether TP73 also affects Pol III transcription of genes controlled by promoters located 5' of the transcription initiation site. To this end, they should monitor the expression of these genes and analyze the assembly of the BRF2-containing form of TFIIIB in response to STAT3 expression. U6 data were used as a negative control for Pol-III genes not regulated by BRF1-containing TFIIIB, but direct data on STAT3-dependent regulation of the U6 gene or other SNAPc-dependent Pol-III genes are lacking.

These are good comments. Yes, we have analyzed the effect of STAT3 or TP73 expression alteration on BRF2 expression by Western blot, and investigating the effect of STAT3 or TP73 depletion on BRF2 assembly at the U6 promoter by performing ChIP assays using the BRF2 antibody. We added these data into Figures3 (C, J and K) and Figures6 (C and I). Thanks very much for the helpful comments.

In addition, co-IP of recombinant TP73 and recombinant TFIIIB subunits (TBP, BDP1, BRF1, BRF2) should be performed to gain better mechanistic insight into which TFIIIB subunit TP73 interacts with in regulating Pol III transcription (presumably BRF1 and BRF2).If the authors are able to address these questions, I am in favor of publication in eLife.

It is a good comment**.** It would be good to look at the interaction between TP73 and BRF1 and BRF2 by performing co-IP assays using recombinant TP73 and recombinant TFIIIB subunits (TBP, BDP1, BRF1, BRF2) in vitro**.** We are really sorry for that we cannot obtain so many recombinant proteins for these assays due to time and fund problems. Since our co-IP assay showed that TP73 did not bind to BRF1 but bind to TBP (Figure 6J); therefore, we still performed co-IP assays using TBP and BRF2 antibodies, and the result was consistent with that obtained in the co-IP assays with TBP and BRF1 antibody. These data were presented in Figure 6—figure supplement 1.

Reviewer #2 (Recommendations for the authors):The Abstract states that "…the pathways and factors that control Pol III-directed transcription have yet to be elucidated." This fails to acknowledge the considerable amount of published work investigating many aspects of pol III transcriptional regulation. The current manuscript makes a very valuable addition, but it is building on a well-established field.

We are sorry for the inaccurate description about the research background. It has been corrected in the Abstract and elswhere in the revised manuscript. Thanks very much for the comment.

P8. The authors should explain what is measured by the CCK8 approach.

Yes, it is a good suggestion; we have added the explanation about the CCK8 method as suggested.

P11. Figure 3I presents ChIP data in which binding of STAT3 is not detected at several pol III-transcribed loci. The conclusion that STAT3 does not act directly at these sites would be greatly strengthened by a positive control confirming that STAT3 binding can clearly be detected at other loci in this assay.

Yes, It is a good suggestion. We added a STAT3-binding positive control ( BCL2 promoter) in Figures 3I.

P16. Figure 6I seems to show some change in levels of BRF1, GTF3C2, and GTF3C3 when TP73 is depleted, contrary to the authors' statement (lines 340-341).

It seems that some change was observed in the levels of those proteins. However, the load control GAPDH has also been showed the same trend. To make these data more confident, we repeated some of the experiments by Western blot and replaced the data in the revised manuscript (Figure 6I).

Petrie et al. (2019) demonstrated that pol III activity promotes the ability of cancer stem cells (CSC) to initiate tumours (Nucleic Acids Res 47, 3937). Given the well-established role of STAT3 in CSC generation, it would be worth discussing if pol III control contributes to the stimulatory effects of STAT3 on CSC.

Yes, it is a good suggestion. We have discussed our findings by referring previous publications about the roles of STAT3 and Pol III products in cell stem cells survival and growth in the Discussion section of the revised manuscript.

Line 362:….STAT3 was originally identified as a positive regulator in RNA polymerase II-directed…should be rephrased into:….STAT3 was originally identified to be a positive regulator in RNA polymerase II-directed…

It has been corrected as suggested. Thanks very much for the helpful comment.

Line 405:Change:….To suppoort this probability…..into….To support this probability…..

We are sorry for the mistake. It has been corrected as suggested.